# EmoGrowth: Incremental Multi-label Emotion Decoding with Augmented Emotional Relation Graph

Kaicheng Fu [* 1]  Changde Du [* 1]  Jie Peng [1 2]  Kunpeng Wang [1 3]  Shuangchen Zhao [1]  Xiaoyu Chen [1 2]
Huiguang He [1 2 3]

## Abstract

Emotion recognition systems face significant challenges in real-world applications, where novel emotion categories continually emerge and multiple emotions often co-occur. This paper introduces multi-label fine-grained class incremental emotion decoding, which aims to develop models capable of incrementally learning new emotion categories while maintaining the ability to recognize multiple concurrent emotions. We propose an **A**ugmented **E**motional **S**emantics **L**earning (AESL) framework to address two critical challenges: past- and future-missing partial label problems. AESL incorporates an augmented **E**motional **R**elation **G**raph (ERG) for reliable soft label generation and affective dimension-based knowledge distillation for future-aware feature learning. We evaluate our approach on three datasets spanning brain activity and multimedia domains, demonstrating its effectiveness in decoding up to 28 fine-grained emotion categories. Results show that AESL significantly outperforms existing methods while effectively mitigating catastrophic forgetting. Our code is available at https://github.com/ChangdeDu/EmoGrowth.

## 1. Introduction

Accurately decoding human emotional states remains a fundamental challenge in affective computing research. While

[*]Equal contribution [1]State Key Laboratory of Brain Cognition and Brain-inspired Intelligence Technology, Institute of Automation, Chinese Academy of Sciences, Beijing, China [2]School of Artificial Intelligence, University of Chinese Academy of Sciences, Beijing, 100049, China [3]School of Biomedical Engineering, ShanghaiTech University, Shanghai, China. Correspondence to: Huiguang He <huiguang.he@ia.ac.cn>.

*Proceedings of the 42^{nd} International Conference on Machine Learning*, Vancouver, Canada. PMLR 267, 2025. Copyright 2025 by the author(s).

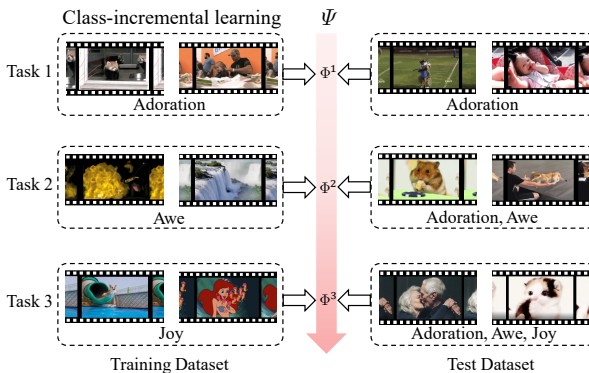

*Figure 1.* The class-incremental learning process is demonstrated through three sequential tasks: Task 1 begins with emotion *adoration*, Task 2 introduces "awe", and Task 3 adds *joy*. The unified model $\psi$ with parameters $\Phi^1$, $\Phi^2$, $\Phi^3$ evolves across tasks while maintaining the ability to recognize all previously learned emotions.

conventional deep learning approaches have demonstrated promising performance (Li & Deng, 2020; Poria et al., 2017), they struggle to adapt to the evolving nature of real-world scenarios, where novel emotion categories continuously emerge to capture increasingly nuanced emotional experiences. As eloquently expressed by the renowned novelist Jeffrey Eugenides (Eugenides, 2003), "*Emotions, in my experience, are not covered by single words. I do not believe in 'sadness', 'joy', or 'regret'*" – this profound observation underscores the inherent complexity of human emotions. Indeed, individuals typically experience a sophisticated blend of multiple emotions simultaneously when responding to emotional stimuli (Fu et al., 2022). Motivated by these insights, we introduce a novel research paradigm: *multi-label fine-grained class incremental emotion decoding*.

As illustrated in Figure 1, multi-label class incremental emotion decoding aims to develop a unified model capable of incrementally learning and integrating knowledge from both existing and emerging emotion classes while comprehensively decoding multiple concurrent emotional states. Unlike traditional **s**ingle-**l**abel **c**lass **i**ncremental **l**earning (SLCIL), the **m**ulti-**l**abel **c**lass **i**ncremental **l**earning (ML-CIL) faces unique challenges in addressing catastrophic forgetting, primarily stemming from past- and future-missing

partial label problems. Consider the past-missing partial label scenario: the left screenshot (*The corgi is diving*) in task 3's training dataset contains the label *Adoration*, yet this emotion remains imperceptible to the model during the current task. Similarly, in the future-missing partial label case, the right screenshot (*The father is combing his daughter's hair*) in task 1's dataset encompasses the label *Joy*, which is inaccessible to the model in the current task. Existing MLCIL approaches either rely on storing historical instances (Kim et al., 2020; Liang & Li, 2022), limiting their practical applicability, or overlook the critical future-missing partial label problem (Du et al., 2022), leading to suboptimal performance.

To address these challenges, we propose a novel **A**ugmented **E**motional **S**emantics **L**earning (AESL) framework. First, we tackle the past-missing partial label problem by introducing an augmented **E**motional **R**elation **G**raph (ERG) module with graph-based label disambiguation. Upon encountering new tasks, this module not only generates reliable soft labels for existing emotion classes but also constructs an enhanced ERG by integrating historical ERG with new data, thereby preserving crucial emotional label correlations. Second, to resolve the future-missing partial label problem, we leverage the affective dimension space – an alternative emotion model capable of representing infinite emotion categories (Russell & Mehrabian, 1977) – to provide complementary domain knowledge (Le et al., 2023) for continuous emotion learning. This insight leads to our development of a relation-based knowledge distillation framework that aligns model features with the affective dimension space. Furthermore, we utilize the ERG to design an emotional semantics learning module incorporating a graph autoencoder, which learns emotion embeddings to facilitate semantic-specific feature decoupling, crucial for enhanced multi-label learning. We conduct extensive evaluations across three datasets: a human brain activity dataset (*Brain27*) with 5 subjects and two multimedia datasets (*Video27* and *Audio28*), implementing multiple incremental learning protocols that encompass up to 28 fine-grained emotion categories. Our key contributions are threefold:

- We pioneer the investigation of multi-label class incremental emotion decoding, advancing emotion recognition capabilities in dynamic real-world environments.

- We develop an innovative augmented emotional semantics learning framework that enhances emotion decoding performance while effectively mitigating catastrophic forgetting in MLCIL scenarios.

- We demonstrate the superior effectiveness of our approach through comprehensive experiments across three datasets and multiple incremental learning protocols.

## 2. Related Work

**Class Incremental Learning.** Class incremental learning has gained significant attention in machine learning research (De Lange et al., 2021; Masana et al., 2020). Traditional methods mainly focus on preventing catastrophic forgetting through regularization (Kirkpatrick et al., 2017), knowledge distillation (Li & Hoiem, 2017), or memory replay (Rebuffi et al., 2017). Recent advances have explored more sophisticated approaches such as parameter isolation (Serra et al., 2018) and dynamic architecture adaptation (Yan et al., 2021).

**Multi-label Class Incremental Learning.** MLCIL addresses the crucial challenge of simultaneously handling incremental class learning and multi-label classification. Recent works have explored various approaches: (Dong et al., 2023) proposed an attention-based knowledge restore and transfer framework, while AGCN (Du et al., 2022) employed GCN for label relationship learning. Online class incremental learning has been addressed through specialized replay buffer designs in PRS (Kim et al., 2020) and OCDM (Liang & Li, 2022). Additional studies have investigated prototype learning (Zhang et al., 2021) and knowledge distillation (Liu et al., 2022) for MLCIL. However, MLCIL specifically focused on emotion decoding remains largely unexplored, particularly concerning the challenges of partial label problems and emotional semantic preservation.

**Class Incremental Emotion Decoding.** The continuous learning of new emotion categories has emerged as a crucial research direction in affective computing. (Churamani & Gunes, 2020) proposed CLIFER, combining a generative model with a complementary learning-based dual-memory model for continual facial expression recognition. (Ma et al., 2022) developed a GCN-based approach for few-shot class-incremental classification across emotion categories, while (Jiménez-Guarneros et al., 2022) introduced weight alignment to address bias in new emotion classes. More recent works have explored adaptive architectures (Wang et al., 2023) and meta-learning approaches (Zhang et al., 2022) for emotion recognition. However, these studies are primarily limited to a small number of coarse-grained emotion categories and fail to address the complexity of human emotional expression.

## 3. Methodology

### 3.1. Problem Formulation

In the MLCIL scenario, we have a sequence of $B$ training tasks $\{\mathcal{D}^1, \mathcal{D}^2, \cdots, \mathcal{D}^B\}$ without overlapping emotion classes, where $\mathcal{D}^b = \{(\mathbf{x}i^b, Yi^b)\}_{i=1}^{n^b}$ is the $b$-th incremental task with $n^b$ training instances. $\mathbf{x}_i^b \in \mathbb{R}^D$ is an instance with classes $Y_i^b \subseteq C^b$. $C^b$ is the label set of task $b$, where

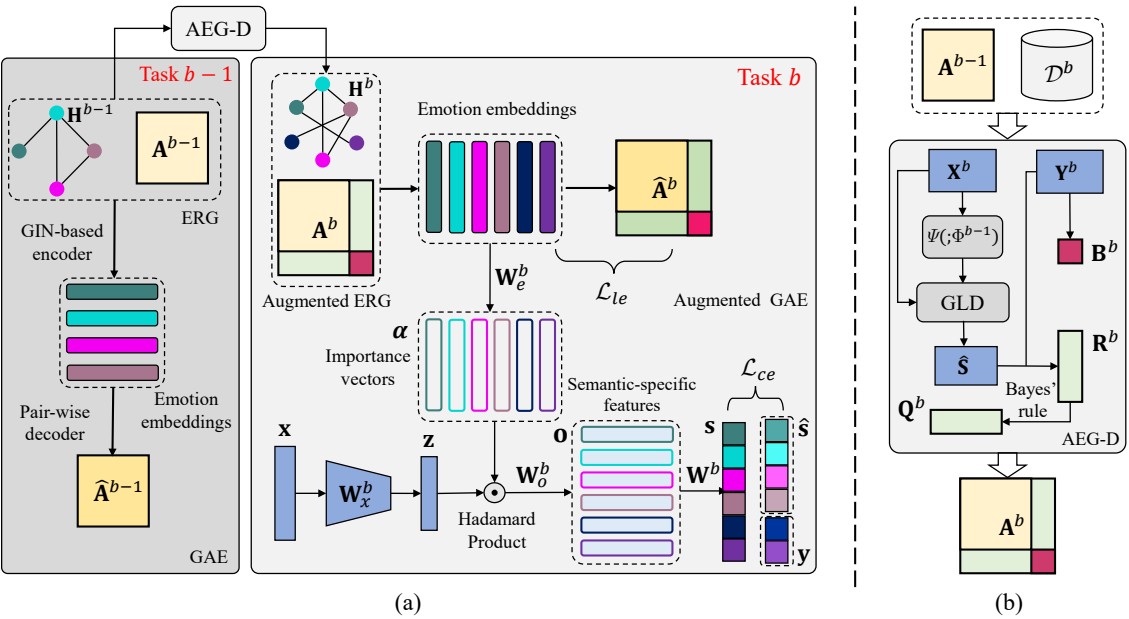

*Figure 2.* The framework of AESL for multi-label class incremental emotion decoding. (a) Emotional semantics learning and semantic-guided feature decoupling procedure in incremental learning scenario. We omit the semantic-guided feature decoupling module in task $b-1$ for clarity. (b) The process of constructing augmented ERG with label disambiguation in task $b$.

$C^b \cap C^{b'} = \varnothing$ for $b \neq b'$. Only data from $\mathcal{D}^b$ is accessible during task $b$ training. $\mathbf{y}_i^b \in \mathbb{R}^{|C^b|}$ is the multi-hot label vector where $y_{ic}^b \in \{0, 1\}$ indicates whether emotion $c$ is relevant to instance $\mathbf{x}_i^b$. After task $b$, the model is evaluated over all seen emotion classes $\mathcal{C}^b = C^1 \cup \cdots C^b$.

### 3.2. Framework Overview

As shown in Figure 2(a), AESL processes each incremental task through four interconnected modules. The ERG module first constructs and maintains the emotion relationship graph, providing the foundation for emotional semantics learning. The graph autoencoder (GAE) then learns emotion embeddings from ERG, which guide the feature decoupling module to extract label-specific features. Finally, relation-based knowledge distillation preserves previously learned knowledge while accommodating new emotion categories. These components work jointly to address both past-missing and future-missing partial label problems in MLCIL.

### 3.3. Augmented Emotional Relation Graph

Here, we first introduce the **A**ugmented **E**motional rela-tion **G**raph module with label **D**isambiguation (AEG-D), as shown in Figure 2(b). At the beginning of task $b$, we have access to new labels $C_b$. Compared with the existing emo-tional relation graph $\mathcal{G}^{b-1}$, we need to augment the node set and adjacency matrix to $V^b$ and $\mathbf{A}^b$, respectively. For the former, we only need to sample $|\mathcal{C}_b|$ vectors from the standard Gaussian distribution. For the latter, it is difficult to infer $\mathbf{A}^b$ directly from statistical label co-occurrence due to the partial label problem. The adjacency matrix on the

given class set $\mathcal{C}$ is defined based on label co-occurrence:

$$\mathbf{A}_{ij} = P(\ell_i \in \mathcal{C} | \ell_j \in \mathcal{C})|_{i \neq j} = \frac{N_{ij}}{N_j}, \quad (1)$$

where $N_{ij}$ is the number of instances with both class $\ell_i$ and $\ell_j$, $N_j$ is the number of instances with class $\ell_j$. When the task $b$ is coming, the augmented adjacency matrix $\mathbf{A}^b$ can be formulated as the following block form (Du et al., 2022):

$$\mathbf{A}^b = \begin{bmatrix} \mathbf{A}^{b-1} & \mathbf{R}^b \\ \mathbf{Q}^b & \mathbf{B}^b \end{bmatrix} \Leftrightarrow \begin{bmatrix} \text{Old-Old} & \text{Old-New} \\ \text{New-Old} & \text{New-New} \end{bmatrix}. \quad (2)$$

$\mathbf{A}^{b-1}$ can be directly inherited from task $b$, and $\mathbf{B}^b$ can be easily computed from $\mathcal{D}^b$. However, $\mathbf{R}^b$ and $\mathbf{Q}^b$ involve the inter-task label relationship between old classes in past tasks and new classes in task $b$. We should first assign soft labels in the past label set $\mathcal{C}^{b-1}$ for the instances in new dataset $\mathcal{D}^b$ for subsequent calculation. For clarity, we use $\mathbf{s} = \psi(\mathbf{x}, \mathbf{A}; \Phi)$ to denote the procedures of emotional semantics learning and semantic-guided feature decoupling, with $\Phi = \{\theta, \phi, \mathbf{W}\}$. Although $\mathbf{s}^b = \psi(\mathbf{x}^b, \mathbf{A}^{b-1}; \Phi^{b-1})$ is a feasible solution for the soft labels construction, this kind of soft labels contain a significant amount of noise and fail to utilize the correlation among instances.

To tackle this problem, we adopt a **G**raph-based **L**abel **D**isambiguation (GLD) module to the label confidence score $\mathbf{s}^b$. Firstly, the similarity between two instances is calculated with Gaussian kernel (omit $b$ without ambiguity) $\mathbf{P}_{ij} = \exp(-\frac{||\mathbf{x}_i - \mathbf{x}_j||^2}{2\sigma^2})$, in which $\mathbf{x}_i$ and $\mathbf{x}_j$ are two different sam-ples in $\mathcal{D}^b$. Following the label propagation procedure, let $\hat{\mathbf{P}} = \mathbf{P}\mathbf{D}^{-1}$ be the propagation matrix by normalizing

weight matrix $\mathbf{P}$ in column, where $\mathbf{D} = \mathrm{diag}[d_1, \cdots, d_{n^b}]$ is the diagonal matrix with $d_j = \sum_{i=1}^{n} \mathbf{P}_{ij}$. Assume that we have access to a past label confidence matrix using $\psi(; \Phi^{b-1})$ for $\mathcal{D}^b$, which denotes as $\mathbf{S} \in \mathbb{R}^{n^b \times |\mathcal{C}^{b-1}|}$. And we set the initial label confidence matrix $\mathbf{F}_0 = \mathbf{S}$. For the $t$-th iteration, the refined label confidence matrix is updated by propagating current labeling confidence over $\hat{\mathbf{P}}$:

$$\mathbf{F}_t = \beta \cdot \hat{\mathbf{P}}^T \mathbf{F}_{t-1} + (1 - \beta) \cdot \mathbf{F}_0. \tag{3}$$

The balancing parameter $\beta \in [0, 1]$ controls the amount of labeling information inherited from iterative label propagation and $\mathbf{F}^0$. Let $\mathbf{F}^*$ be the final label confidence matrix and also serve as the soft labels after disambiguation, which means $\hat{\mathbf{S}} = \mathbf{F}^*$. We set the balancing parameter $\beta$ to 0.95 during label disambiguation according to (Chen et al., 2020).

With the dataset $\mathcal{D}^b$ and soft label matrix $\hat{\mathbf{S}}$, we are able to compute $\mathbf{R}^b \in \mathbb{R}^{|\mathcal{C}^{b-1}| \times |\mathcal{C}^b|}$ as follows:

$$\mathbf{R}_{ij}^b = P(\ell_i \in \mathcal{C}^{b-1} | \ell_j \in C^b) = \frac{\sum_{\mathbf{x}} \hat{s}_i y_j}{N_j}, \tag{4}$$

in which $\hat{s}_i$ denotes the value of class $i$ corresponding to instance $\mathbf{x}$ in the soft label matrix, and $y_j$ refers to the value of class $j$ corresponding to the same instance in the label matrix derived from $\mathcal{D}^b$. Naturally, following the Bayes' rule, we can obtain the $\mathbf{Q}^b \in \mathbb{R}^{|C^b| \times |\mathcal{C}^{b-1}|}$ by:

$$\begin{aligned}
\mathbf{Q}_{ji}^b &= P(\ell_j \in C^b | \ell_i \in \mathcal{C}^{b-1}) \\
&= \frac{P(\ell_i \in \mathcal{C}^{b-1} | \ell_j \in C^b) P(\ell_j \in C^b)}{P(\ell_i \in \mathcal{C}^{b-1})} = \frac{\mathbf{R}_{ij}^b N_j}{\sum_{\mathbf{x}} \hat{s}_i}.
\end{aligned} \tag{5}$$

Above all, we have constructed the adjacency matrix $\mathbf{A}^b$ and achieved continual learning of new emotion categories in the multi-label scenario. It is noticeable that, in our experiments, we actually utilize a symmetric adjacency matrix for model training by applying the $\frac{\mathbf{A} + \mathbf{A}^T}{2}$ operation.

### 3.4. Emotional Semantics Learning

Now, we focus on how to obtain the emotion embeddings in task $b$. We construct an augmented ERG $\mathcal{G}^b = (V^b, E^b)$, where $V^b$ represents nodes corresponding to class labels $\mathcal{C}^b$ and $E^b$ refers to edges. Then, we adopt a GAE to project emotion labels into a label co-occurrence semantic space with the ERG. We exploit Graph Isomorphism Network (GIN) (Xu et al., 2019) as the encoder of our GAE due to its powerful representation learning capability. Specifically, given a feature matrix of nodes $\mathbf{H}_l^b \in \mathbb{R}^{|\mathcal{C}^b| \times d_l}$ in which each row refers to the embedding of an emotion label and $d_l$ corresponds to the dimensionality of node features in the $l$-th GIN layer, the node features are able to update within a GIN layer with a message passing strategy by:

$$\mathbf{H}_{l+1}^b = f_{l+1}[(1 + \epsilon_{l+1}) \mathbf{H}_l^b + \mathbf{A}^b \mathbf{H}_l^b; \theta_{l+1}^b], \tag{6}$$

in which $\mathbf{H}_{l+1}^b \in \mathbb{R}^{|\mathcal{C}^b| \times d_{l+1}}$ is the updated feature matrix of nodes, $f_{l+1}(:, \theta_{l+1})$ refers to a fully-connected neural network. Additionally, $\epsilon_{l+1}$ is a learnable parameter which regulates the importance of the node's own features during the process of neighborhood aggregation. Unlike object labels in image classification (Chen et al., 2019), emotion category labels are difficult to obtain initial word embeddings directly from language models. Consequently, the initial feature matrix of nodes $\mathbf{H}_0^b \in \mathbb{R}^{|\mathcal{C}^b| \times d_0}$ is initialized by standard Gaussian distribution and we set $d_0 = |\mathcal{C}^b|$ (can also be other task-agnostic constants). After stacking $L$ GIN layers, we use $\mathbf{E}^b = \mathbf{H}_L^b \in \mathbb{R}^{|\mathcal{C}^b| \times d_L}$ as the final emotion label semantic embeddings in task $b$ for further semantic-specific feature extraction.

Furthermore, we introduce a pairwise decoder to reconstruct the adjacency matrix $\mathbf{A}^b$, which can ensure that the obtained label embeddings capture the topological structure of the label semantic space well. The loss function of the pairwise decoder can be written as:

$$\mathcal{L}_{le} = \frac{1}{|\mathcal{C}^b|^2} \sum_{i=1}^{|\mathcal{C}^b|} \sum_{j=1}^{|\mathcal{C}^b|} [\frac{(\mathbf{e}_i^b - \bar{\mathbf{e}}^b)^T (\mathbf{e}_j^b - \bar{\mathbf{e}}^b)}{||\mathbf{e}_i^b - \bar{\mathbf{e}}^b|| ||\mathbf{e}_j^b - \bar{\mathbf{e}}^b||} - \hat{\mathbf{A}}_{ij}^b]^2, \tag{7}$$

where $\bar{\mathbf{e}}^b = \mathbb{E}_i[\mathbf{e}_i^b]$ corresponds to the average emotion embeddings in task $b$, $\mathbf{e}_i^b$ denotes the $i$-th row of $\mathbf{E}^b$, and $\hat{\mathbf{A}}^b = \mathbf{A}^b + \mathbf{I}^b$, with $\mathbf{I}^b$ being an identity matrix. Overall, emotional semantics learning aims to fit the function $f(\cdot, \theta^b)$, leading to $\mathbf{E}^b = f(\mathbf{H}_0^b, \mathbf{A}^b; \theta^b)$ in task $b$.

### 3.5. Semantic-Guided Feature Decoupling

A multi-label classification model with semantic-guided feature decoupling can be regarded as the composition of a semantic-specific feature extractor $g$ and a classification head $\mathbf{W}$ (omitting bias for simplicity), where $g(; \phi^b)$ : $\mathbb{R}^D \times \mathbb{R}^{|\mathcal{C}^b| \times d_L} \to \mathbb{R}^{|\mathcal{C}^b| \times d}$ and $\mathbf{W}^b \in \mathbb{R}^{d \times |\mathcal{C}^b|}$ in task $b$. For a specific emotion label $\ell_k \in \mathcal{C}^b$, the semantic-specific mapping $g_k$ can be formulated as $g_k(\mathbf{x}^b, \mathbf{e}_k^b; \phi^b) \in \mathbb{R}^d$. The linear layer can be further decomposed into the combination of classifiers $\mathbf{W}^b = [\mathbf{w}_1, \cdots, \mathbf{w}_{|\mathcal{C}^b|}]$, in which each classifier corresponds to one emotion embedding. The classification head will be expanded for new classes as the continual learning progresses. The key issue is to design the semantic-specific feature extractor $g_k$.

To achieve this, we first map the instance representation $\mathbf{x}^b$ from the original feature space to a more powerful deep latent feature $\mathbf{z}$ with a fully-connected network having parameters $\mathbf{W}_x^b \in \mathbb{R}^{D \times d_z}$ and $\mathbf{b}_x^b \in \mathbb{R}^{d_z}$. In order to utilize emotional semantics to guide the feature extraction for each label, we adopt an attention-like mechanism. Concretely, we attempt to obtain feature importance values $\boldsymbol{\alpha}_k$ for each emotion category by using a fully-connected network followed by a sigmoid function for $\mathbf{e}_k^b$ with parameters

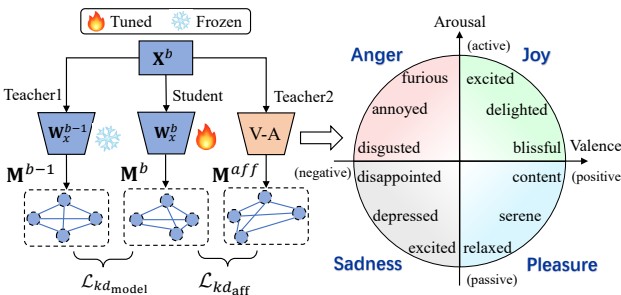

*Figure 3.* Diagram of relation-based knowledge distillation with two teachers in the process of training task $b$. Teachers 1&2 were frozen during training. Each discrete emotion category represents a point in the affective space formed by Arousal and Valence.

$\mathbf{W}_e^b \in \mathbb{R}^{d_L \times d_z}$ and $\mathbf{b}_e^b \in \mathbb{R}^{d_z}$. Then, we select pertinent features for each emotion category via the Hadamard product between the feature importance vector and the latent representation. Successively, we can obtain the semantic-specific feature for each emotion category from another fully-connected network. This procedure can be formulated as follows:

$$\mathbf{o}_k = \zeta[\mathbf{W}_o^{b^T}(\mathbf{z} \odot \boldsymbol{\alpha}_k) + \mathbf{b}_o^b], \tag{8}$$

where $\mathbf{W}_o^b \in \mathbb{R}^{d_z \times d}$ and $\mathbf{b}_o^b \in \mathbb{R}^d$ are shared learnable parameters. $\odot$ refers to the Hadamard product, and $\zeta$ denotes the activation function. At this point, we have defined the semantic-specific feature extractor $\mathbf{o}_k = g_k(\mathbf{x}^b, \mathbf{e}_k^b; \phi^b)$, in which $\phi = \{\mathbf{W}_x, \mathbf{W}_e, \mathbf{W}_o, \mathbf{b}_x, \mathbf{b}_e, \mathbf{b}_o\}$. Then, we can predict the confidence score of the presence of emotion label $\ell_k$ through the corresponding classifier:

$$s_k = \sigma(\mathbf{w}_k^T \mathbf{o}_k + b_k) = \sigma(\mathbf{w}_k^T g_k(\mathbf{x}^b, \mathbf{e}_k^b; \phi^b) + b_k).$$
$$k \in \{1, \cdots, |\mathcal{C}^b|\} \tag{9}$$

### 3.6. Relation-based Knowledge Distillation

Although we have achieved the semantic-specific feature learning and overcome the past-missing partial label problem by AEG-D, we have not yet addressed the issue of future-missing partial label problem in MLCIL. Previous studies (Schlosberg, 1954; Russell & Mehrabian, 1977) have shown that the affective dimension, as a complementary emotion model to emotion category, can represent infinitely many emotion categories within its constructed affective space. We propose that incorporating the domain knowledge of affective dimension space into the model can alleviate the problem of future-missing partial label problem. During the training process for each task, we attempt to align the feature space of our model with the predefined affective space constructed by some affective dimensions such as *Arousal* and *Valence*. Taking into account the heterogeneity of the two spaces, we adopt **r**elation-based **k**nowledge **d**istillation (RKD). Specifically, we firstly calculate the **r**epresentation

similarity **m**atrix (RSM) (Kriegeskorte et al., 2008) obtained from model feature $\mathbf{z}$ for task $b$:

$$\mathbf{M}_{ij}^b = \frac{(\mathbf{z}_i - \bar{\mathbf{z}})^T(\mathbf{z}_j - \bar{\mathbf{z}})}{||\mathbf{z}_i - \bar{\mathbf{z}}||||\mathbf{z}_j - \bar{\mathbf{z}}||}, \tag{10}$$

where $\bar{\mathbf{z}} = \mathbb{E}_i[\mathbf{z}_i]$ denotes the mean model feature of all instances. Similarly, RSM obtained from affective dimension feature $\boldsymbol{\tau}$ can be formulated as:

$$\mathbf{M}_{ij}^{\text{aff}} = \frac{(\boldsymbol{\tau}_i - \bar{\boldsymbol{\tau}})^T(\boldsymbol{\tau}_j - \bar{\boldsymbol{\tau}})}{||\boldsymbol{\tau}_i - \bar{\boldsymbol{\tau}}||||\boldsymbol{\tau}_j - \bar{\boldsymbol{\tau}}||}. \tag{11}$$

Then, we define the similarity loss $\mathcal{L}_{kd_{\text{aff}}}$ as:

$$\begin{aligned}\mathcal{L}_{kd_{\text{aff}}} &= \mathbb{E}_{i \neq j}[\mathcal{L}_{kd_{\text{aff}}}^{ij}] \\ &= \mathbb{E}_{i \neq j}\left\{[\text{arctanh}(\mathbf{M}_{ij}^b) - \text{arctanh}(\mathbf{M}_{ij}^{\text{aff}})]^2\right\},\end{aligned} \tag{12}$$

where $\mathcal{L}_{kd_{\text{aff}}}^{ij}$ is the sample-based centered kernel alignment index. We leverage $\text{arctanh}$ to reparameterize the similarity values from the interval $(-1, 1)$ to $(-\infty, \infty)$ to approximately obey a Gaussian distribution. Besides, to ensure training stability, we simultaneously pull together $\mathbf{M}_{ij}^b$ and $\mathbf{M}_{ij}^{b-1}$ using the same method to obtain $\mathcal{L}_{kd_{\text{model}}}$. In this way, the model has access to two teachers: the affective dimension and the old model, as shown in Figure 3. This approach derives the overall knowledge distillation loss:

$$\mathcal{L}_{kd} = \lambda_1 \mathcal{L}_{kd_{\text{model}}} + \lambda_2 \mathcal{L}_{kd_{\text{aff}}}. \tag{13}$$

### 3.7. Objective Function

As mentioned above, the prediction confidence scores $\mathbf{s}$ for an instance $\mathbf{x}$ can be computed by Eq. 9, which denotes $\mathbf{s} = [s_1, \cdots, s_{|\mathcal{C}^b|}]^T \in \mathbb{R}^{|\mathcal{C}^b|}$ in task $b$. We have access to the ground truth from $\mathcal{D}^b$, which is denoted as the multi-hot vector $\mathbf{y} = [y_1, \cdots, y_{C^b}]^T \in \mathbb{R}^{|C^b|}$. Additionally, we have computed the soft labels for the previous emotion categories using the model $b - 1$, as described in Section 3.3, which are denoted as $\hat{\mathbf{s}} = [\hat{s}_1, \cdots, \hat{s}_{|\mathcal{C}^{b-1}|}]^T \in \mathbb{R}^{|\mathcal{C}^{b-1}|}$, and $C^b \cup \mathcal{C}^{b-1} = \mathcal{C}^b$. In summary, we train task $b$ using the mixed ground truth $\tilde{\mathbf{y}} = [\hat{\mathbf{s}}^T, \mathbf{y}^T]^T \in \mathbb{R}^{|\mathcal{C}^b|}$, with a the binary cross entropy loss, formulated as follows:

$$\mathcal{L}_{ce} = -\sum_{i=1}^{|\mathcal{C}^b|}[\tilde{y}_i \log(s_i) + (1 - \tilde{y}_i)\log(1 - s_i)]. \tag{14}$$

Finally, our model is trained with the following objective function in an end-to-end manner:

$$\mathcal{L} = \mathcal{L}_{ce} + \lambda_1 \mathcal{L}_{kd_{\text{model}}} + \lambda_2 \mathcal{L}_{kd_{\text{aff}}} + \lambda_3 \mathcal{L}_{le}. \tag{15}$$

After training in task $b$, given an unseen instance, its associated label set is predicted as $\{\ell_k | s_k > 0.5, 1 \leq k \leq \mathcal{C}^b\}$. The algorithm of AESL is written in Appendix A.

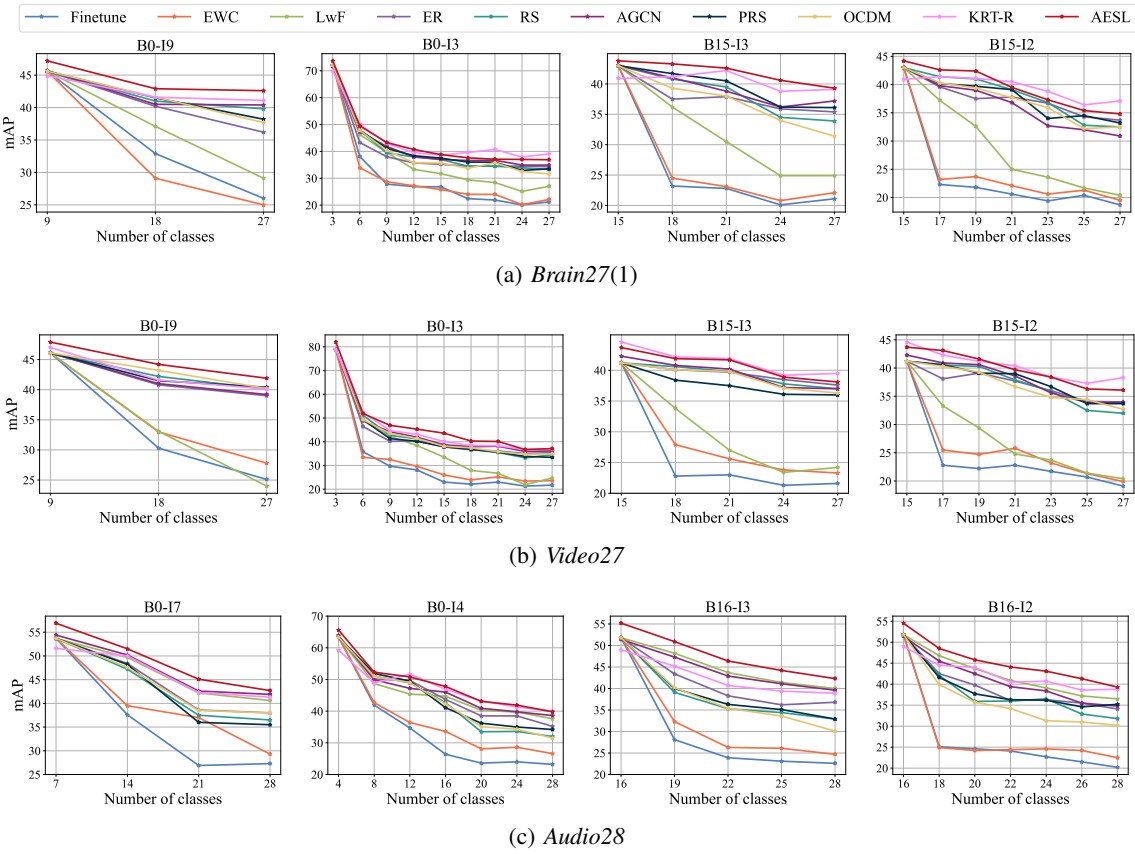

(a) *Brain27*(1)

(b) *Video27*

(c) *Audio28*

*Figure 4.* Comparison results (mAP) on three datasets used in our experiment under different protocols against compared CIL methods.

## 4. Experiments

### 4.1. Experimental Setup

**Datasets.** For thoroughly evaluating the performance of AESL and comparing approaches, three datasets are leveraged for experimental studies including *Brain27* (Horikawa et al., 2020), *Video27* (Cowen & Keltner, 2017) and *Audio28* (Cowen et al., 2020). We evaluate our method using two popular protocols in class incremental learning work (Dong et al., 2023), including (1) training all emotion classes in several splits and (2) first training a base model on a few classes while the remaining classes being divided into several tasks. For *Brain27* and *Video27*, we split the datasets with B0-I9 (base class is 0 and incremental class is 9), B0-I3, B15-I3 and B15-I2. For *Audio28*, we split the dataset with B0-I7, B0-I4, B16-I3 and B16-I2.

**Baselines.** The performance of AESL is compared with multiple essential and state-of-art class incremental methods. *Finetune* is a baseline which means fine-tuning the model without any anti-forgetting constraints. We select four SLCIL methods including *EWC* (Kirkpatrick et al., 2017), *LwF* (Lee et al., 2019), *ER* (Rolnick et al., 2019) and *RS* (Vitter, 1985) for comparison. Furthermore, other

three well-established MLCIL approaches *AGCN* (Du et al., 2022), *PRS* (Kim et al., 2020), *OCDM* (Liang & Li, 2022), and *KRT-R* (Dong et al., 2023) are also employed as comparing approaches. Besides, we set the *Upper-bound* as the supervised training on the data of all tasks. More details about the experimental setups can be found in Appendix B.

### 4.2. Experimental Results

**Comparative Studies.** Tables 1, 2, and 3 show the results on subject 1 of *Brain27* (more results are shown in Appendix C), *Video27* and *Audio*, respectively. We can observe that AESL shows obvious superiority under different datasets and protocols, in terms of three widely used metrics mAP, maF1 and miF1 (Zhang & Zhou, 2013). Especially for *Brain27*, AESL has a relative improvement of 9.6% in mAP (4 protocols averaged) and 9.7% in maF1 compared with the second place method. Figure 4 exhibits the comparison curves of AESL and comparing methods, which indicates that our method is consistently optimal at each task of incremental learning. Among the compared methods, EWC and LwF are traditional SLCIL methods. We can find that EWC is not suitable for direct application to MLCIL task due to its poor performance. In contrast, LwF achieves impressive per-

*Table 1.* Class incremental results on subject 1 of *Brain27* dataset. AGCN, PRS, OCDM and KRT-R are MLCIL algorithms among these compared methods.

| Method | Brain27 B0-I9 | | | | Brain27 B0-I3 | | | | Brain27 B15-I3 | | | | Brain27 B15-I2 | | | |
|---|---|---|---|---|---|---|---|---|---|---|---|---|---|---|---|---|
| | Avg. Acc | Last Acc | | | Avg. Acc | Last Acc | | | Avg. Acc | Last Acc | | | Avg. Acc | Last Acc | | |
| | mAP | maF1 | miF1 | mAP | mAP | maF1 | miF1 | mAP | mAP | maF1 | miF1 | mAP | mAP | maF1 | miF1 | mAP |
| Upper-bound | - | 39.1 | 47.2 | 45.8 | - | 39.1 | 47.2 | 45.8 | - | 39.1 | 47.2 | 45.8 | - | 39.1 | 47.2 | 45.8 |
| Finetune | 34.8 | 9.2 | 19.3 | 26.0 | 30.8 | 5.0 | 13.8 | 21.2 | 26.0 | 5.0 | 13.9 | 21.1 | 23.7 | 3.6 | 13.2 | 18.7 |
| EWC | 33.2 | 8.1 | 17.1 | 25.0 | 30.9 | 5.0 | 13.9 | 22.1 | 26.7 | 5.3 | 14.3 | 22.1 | 24.8 | 3.6 | 13.2 | 19.5 |
| LwF | 37.3 | 12.2 | 29.1 | 29.1 | 37.0 | 23.4 | 40.0 | 27.0 | 31.9 | 14.2 | 28.6 | 24.9 | 29.1 | 15.2 | 31.8 | 20.4 |
| ER | 40.7 | 8.0 | 11.7 | 36.2 | 40.2 | 4.9 | 9.1 | 34.3 | 37.9 | 9.7 | 12.3 | 35.4 | 37.5 | 9.6 | 11.7 | 33.7 |
| RS | 42.1 | 9.4 | 12.6 | 39.8 | 41.2 | 4.5 | 7.4 | 33.3 | 38.3 | 8.0 | 11.1 | 33.9 | 37.9 | 5.1 | 8.7 | 32.5 |
| AGCN | 42.2 | 29.5 | 44.5 | 40.4 | 42.1 | 35.4 | 43.7 | 34.9 | 39.3 | 28.8 | **41.7** | 37.2 | 36.3 | 24.4 | 36.4 | 30.9 |
| PRS | 41.6 | 9.3 | 15.1 | 38.2 | 41.7 | 5.5 | 8.4 | 33.5 | 39.5 | 8.2 | 12.2 | 36.1 | 37.7 | 9.4 | 13.1 | 33.2 |
| OCDM | 41.6 | 9.7 | 15.9 | 37.7 | 40.6 | 5.3 | 7.6 | 31.6 | 37.2 | 4.9 | 7.1 | 31.4 | 37.3 | 4.4 | 7.6 | 32.5 |
| KRT-R | 42.5 | 18.2 | 30.3 | 41.1 | **44.3** | 22.9 | 32.1 | **39.1** | 40.4 | 20.0 | 33.2 | 39.1 | **39.5** | 20.3 | 33.7 | **37.1** |
| **AESL** | **44.2** | **32.8** | **44.7** | **42.6** | 43.8 | **37.1** | **44.0** | 36.9 | **41.9** | **32.5** | 41.7 | **39.3** | **39.5** | **26.8** | **36.5** | 34.8 |

*Table 2.* Class incremental results on *Video27* dataset. AGCN, PRS, OCDM, and KRT-R are MLCIL algorithms among these compared methods.

| Method | Video27 B0-I9 | | | | Video27 B0-I3 | | | | Video27 B15-I3 | | | | Video27 B15-I2 | | | |
|---|---|---|---|---|---|---|---|---|---|---|---|---|---|---|---|---|
| | Avg. Acc | Last Acc | | | Avg. Acc | Last Acc | | | Avg. Acc | Last Acc | | | Avg. Acc | Last Acc | | |
| | mAP | maF1 | miF1 | mAP | mAP | maF1 | miF1 | mAP | mAP | maF1 | miF1 | mAP | mAP | maF1 | miF1 | mAP |
| Upper-bound | - | 36.8 | 46.3 | 45.4 | - | 36.8 | 46.3 | 45.4 | - | 36.8 | 46.3 | 45.4 | - | 36.8 | 46.3 | 45.4 |
| Finetune | 33.8 | 6.6 | 13.7 | 25.1 | 31.5 | 4.2 | 13.0 | 21.7 | 26.0 | 4.3 | 13.3 | 21.6 | 24.3 | 4.2 | 13.7 | 19.1 |
| EWC | 35.6 | 7.5 | 17.8 | 27.8 | 32.9 | 4.7 | 13.2 | 23.6 | 28.4 | 4.9 | 13.9 | 23.3 | 26.0 | 3.9 | 13.1 | 19.9 |
| LwF | 34.4 | 6.8 | 23.3 | 24.0 | 38.2 | 19.9 | 37.3 | 24.7 | 30.0 | 12.5 | 33.4 | 24.2 | 27.7 | 15.8 | 32.7 | 20.4 |
| ER | 42.0 | 5.1 | 7.0 | 39.0 | 43.0 | 4.5 | 4.8 | 35.4 | 39.4 | 8.0 | 8.3 | 37.6 | 37.2 | 10.1 | 12.6 | 34.0 |
| RS | 42.8 | 4.6 | 6.1 | 40.1 | 43.6 | 4.6 | 7.8 | 34.5 | 39.3 | 4.2 | 5.2 | 37.0 | 37.2 | 6.7 | 9.9 | 32.0 |
| AGCN | 42.1 | 22.4 | 39.4 | 39.2 | 44.5 | 34.2 | 44.5 | 36.1 | 39.5 | 22.7 | 38.4 | 37.0 | 38.0 | 23.8 | 36.2 | 34.0 |
| PRS | 42.6 | 9.5 | 15.0 | 40.4 | 43.0 | 5.8 | 9.6 | 33.4 | 37.8 | 8.9 | 13.4 | 36.0 | 37.7 | 7.4 | 13.3 | 33.7 |
| OCDM | 43.1 | 5.5 | 6.8 | 40.2 | 43.8 | 5.0 | 7.8 | 35.0 | 38.9 | 4.9 | 6.6 | 36.3 | 37.1 | 5.2 | 5.8 | 32.7 |
| KRT-R | 42.9 | 26.7 | 35.8 | 40.1 | 45.5 | 26.3 | 34.7 | 37.0 | **41.5** | **25.0** | 35.5 | **40.4** | 39.5 | 24.2 | 34.2 | **38.3** |
| **AESL** | **44.6** | 23.4 | **39.7** | **41.9** | **47.1** | **35.2** | **45.0** | **37.1** | **41.5** | 23.5 | **39.2** | 38.1 | **39.8** | **24.5** | **36.7** | 36.1 |

*Table 3.* Class incremental results on *Audio28* dataset. AGCN, PRS, OCDM and KRT-R are MLCIL algorithms among these compared methods.

| Method | Audio28 B0-I7 | | | | Audio28 B0-I4 | | | | Audio28 B16-I3 | | | | Audio28 B16-I2 | | | |
|---|---|---|---|---|---|---|---|---|---|---|---|---|---|---|---|---|
| | Avg. Acc | Last Acc | | | Avg. Acc | Last Acc | | | Avg. Acc | Last Acc | | | Avg. Acc | Last Acc | | |
| | mAP | maF1 | miF1 | mAP | mAP | maF1 | miF1 | mAP | mAP | maF1 | miF1 | mAP | mAP | maF1 | miF1 | mAP |
| Upper-bound | - | 51.4 | 61.1 | 57.1 | - | 51.4 | 61.1 | 57.1 | - | 51.4 | 61.1 | 57.1 | - | 51.4 | 61.1 | 57.1 |
| Finetune | 36.4 | 9.2 | 14.8 | 27.3 | 33.9 | 5.3 | 10.0 | 23.3 | 29.9 | 4.4 | 10.3 | 22.6 | 27.6 | 2.8 | 8.2 | 20.2 |
| EWC | 37.9 | 8.3 | 14.3 | 29.3 | 37.1 | 5.4 | 10.5 | 26.6 | 32.2 | 4.4 | 9.7 | 24.7 | 28.1 | 2.8 | 8.7 | 22.5 |
| LwF | 46.6 | 37.9 | 51.7 | 40.6 | 45.8 | 39.9 | 49.8 | 37.6 | 45.0 | **32.3** | 45.2 | 40.0 | 42.3 | 28.8 | 41.4 | 36.5 |
| ER | 44.7 | 8.1 | 11.4 | 38.0 | 45.6 | 6.5 | 5.5 | 35.2 | 41.3 | 9.2 | 13.3 | 36.8 | 39.4 | 10.1 | 13.6 | 34.1 |
| RS | 43.7 | 8.1 | 12.3 | 36.5 | 43.6 | 5.9 | 9.3 | 32.0 | 38.7 | 7.5 | 11.7 | 32.9 | 38.2 | 5.8 | 11.6 | 31.8 |
| AGCN | 47.3 | 35.3 | 50.9 | 41.9 | 46.6 | 37.5 | 51.0 | 38.6 | 44.5 | 29.3 | 44.6 | 39.6 | 41.1 | 27.5 | 42.4 | 34.8 |
| PRS | 43.3 | 9.0 | 12.8 | 35.5 | 44.5 | 6.8 | 9.0 | 34.2 | 39.2 | 6.2 | 8.3 | 32.9 | 39.1 | 8.8 | 11.4 | 35.2 |
| OCDM | 44.5 | 8.7 | 12.0 | 38.0 | 43.8 | 7.5 | 8.8 | 31.5 | 38.2 | 5.5 | 9.7 | 30.1 | 36.3 | 3.7 | 7.9 | 30.2 |
| KRT-R | 46.3 | 8.2 | 23.7 | 41.3 | 47.3 | 18.1 | 33.0 | **40.0** | 42.6 | 11.4 | 27.9 | 38.9 | 42.3 | 13.1 | 29.2 | 38.8 |
| **AESL** | **49.0** | **38.4** | **51.8** | **42.7** | **48.7** | **41.1** | **51.7** | 39.8 | **47.8** | 32.3 | **48.0** | **42.3** | **45.3** | **30.8** | **45.1** | **39.3** |

formance on the *Audio28* dataset. We infer that knowledge distillation, which essentially provides soft labels for old classes, contributes to overcoming catastrophic forgetting in MLCIL. Besides, it is noticeable that rehearsal-based methods, including ER, RS, PRS, OCDM and KRT-R, are not satisfactory, especially for maF1 and miF1. This is because just saving the labels of current task aggravates the partial label problem in subsequent training, which illustrates that data replay is not suitable for MLCIL. Furthermore, AGCN serves as a strong baseline which can rank 2nd for most cases, and our method surpasses AGCN by incorporating effective label semantics learning and introducing knowledge distillation from the affective space.

Furthermore, we adopt the Friedman test (Demšar, 2006) for statistical testing in order to discuss the relative performance among the compared methods[1]. If there are $k$ algorithms and $N$ datasets, we take use of the average ranks of algorithms $R_j = \frac{1}{N} \sum_i r_i^j$ for Friedman test in which $r_i^j$ is the ranks of the $j$-th algorithm on the $i$-th dataset. If the null-hypothesis is that all the algorithms have the equivalent performance, the Friedman statistic $F_F$ which will satisfy

---

[1]We average the last mAP of four protocols regarding each dataset for further analysis.

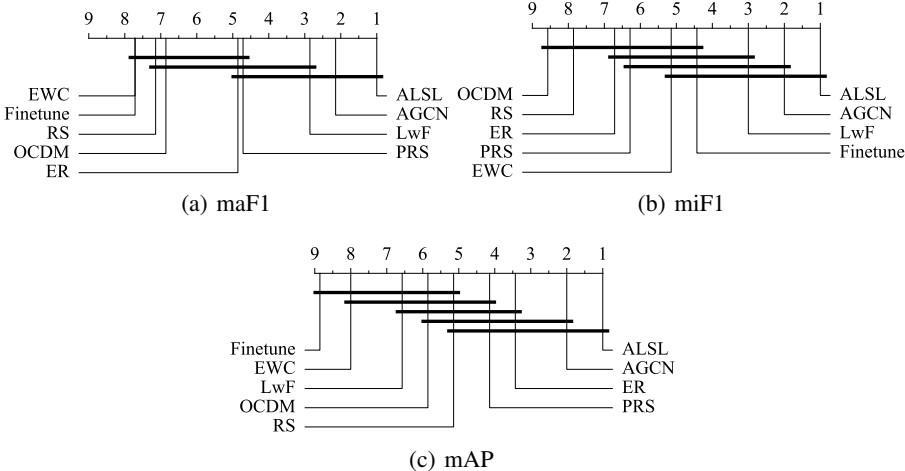

(a) maF1

(b) miF1

(c) mAP

*Figure 5.* Pairwise comparisons with the Nemenyi test in 7 datasets and 9 algorithms used in our experiments. Algorithms not connected with each other in the CD diagram are considered to have significantly different performance ($CD = 4.540$ at 0.05 significance level).

the F-distribution with $k - 1$ and $(k - 1)(N - 1)$ degrees of freedom can be written as:

$$F_F = \frac{(N - 1)\chi_F^2}{N(k - 1) - \chi_F^2}, \qquad (16)$$

in which

$$\chi_F^2 = \frac{12N}{k(k + 1)} \left[ \sum_{j=1}^{k} R_j^2 - \frac{k(k + 1)^2}{4} \right]. \qquad (17)$$

Table 4 shows the Friedman statistics $F_F$ and the corresponding critical value in regard to each metric (# comparing algorithms $k = 9$, # datasets $N = 7$). With respect to each metric, the null hypothesis of equivalent performance among the compared methods can be rejected at the 0.05 significance level.

Then, we perform the strict post-hoc Nemenyi test (Demšar, 2006) which is used to account for pairwise comparisons for all compared approaches. The critical difference (CD) value of the rank difference between two algorithms is:

$$CD = q_\alpha \sqrt{\frac{k(k + 1)}{6N}}, \qquad (18)$$

in which $q_\alpha = 3.102$ at 0.05 significance level. Therefore, one algorithm can be considered as having significantly different performance than another method if their average ranks difference is larger than CD ($CD = 4.540$ in our experimental setting). Figure 5 reports the CD diagrams on each metric, where the average rank of each compared method is marked along the axis (the smaller the better). The algorithms that are not connected by a horizontal line are considered to have significant differences in performance. We can observe that: (1) In terms of mAP,

*Table 4.* Friedman statistics $F_F$ in terms of each metric and the critical value at 0.05 significance level. (# compared algorithms $k = 9$, # subjects $N = 7$.)

| maF1 | miF1 | mAP | critical value |
|---|---|---|---|
| 37.025 | 64.000 | 70.696 | 2.138 |

*Table 5.* The contribution of each component. Accuracy of these models is measured by mAP.

| Model | ESL | LD | RKD | Avg. Acc | Δ | Last Acc | Δ |
|---|---|---|---|---|---|---|---|
| w/o ESL&LD | | | | 47.6 | -1.4 | 41.3 | -1.4 |
| +SE | | | | 45.3 | -3.7 | 40.9 | -1.8 |
| +LE | | | ✓ | 46.8 | -2.2 | 41.9 | -0.8 |
| +AD | | | | 48.4 | -0.6 | 42.3 | -0.4 |
| w/o LD | ✓ | | ✓ | 48.1 | -0.9 | 42.0 | -0.7 |
| w/o RKD | ✓ | ✓ | | 48.3 | -0.7 | 42.1 | -0.6 |
| +LR | ✓ | ✓ | | 47.0 | -2.0 | 41.8 | -0.9 |
| **AESL** | ✓ | ✓ | ✓ | **49.0** | 0.0 | **42.7** | 0.0 |

AESL and other MLCIL approaches (except OCDM) significantly outperform SLCIL methods. (2) In terms of miF1 and maF1, rehearsal-free methods are significantly better than rehearsal-based algorithms. (3) AESL is not significantly different from some MLCIL approaches. This is due to the factor that these approaches beat other comparing approaches and the Nemenyi test fails to detect that AESL achieves a consistently better average ranks than other methods on all evaluation metrics.

**Ablation Studies.** To evaluate the roles of AESL's three key components, we conduct ablation experiments on the *Audio28* dataset under the B0-I7 setting. We design seven baselines, as shown in Table 5: (1) **w/o ESL&LD**: Feed the feature vector **z** directly into the classifier without semantic-guided feature decoupling. (2) **w/o ESL&LD, +SE**: Extract sentence embeddings of emotion category descriptions via

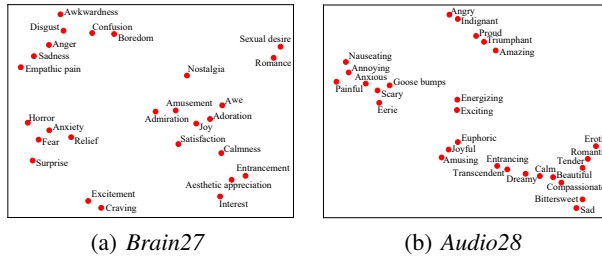

(a) *Brain27*      (b) *Audio28*

*Figure 6.* t-SNE on the emotion embeddings learned by AESL.

the advanced LLaMA 3.1-8B (Dubey et al., 2024) and utilize them as emotion embeddings. (3) **w/o ESL&LD, +LE**: Assign a learnable embedding to each new emotion category. (4) **w/o ESL&LD, +AD**: Directly integrate the sample-wise affective dimension into category-wise emotion embeddings. (5) **w/o LD**: Use original confidence score **S** for constructing adjacency matrix without label disambiguation. (6) **w/o RKD**: Remove the module of knowledge distillation from affective space. (7) **w/o RKD, + LR**: Replace RKD with a linear regressor with a non-linear activation to predict the target affective dimension, thereby constraining **z** and incorporating the affective dimension.

It is noteworthy, but not surprising, that even when sentence embeddings are extracted using state-of-the-art large language models, their usage as emotion category labels fails to achieve comparable performance to semantic-guided feature decoupling. This performance degradation suggests that the effectiveness of AESL relies not only on the quality of embeddings but also on the explicit structural alignment provided by semantic-guided decoupling. Results show that all the three components in AESL are critical for preventing forgetting and improving the model performance of MLCIL.

**Emotional Semantics Visualization.** Emotional semantics learning is an essential module for our approach. In Figure 6, we adopt the t-SNE (Van der Maaten & Hinton, 2008) to visualize the emotion embeddings learned by the emotional semantics learning module. It is clear to see that, the learned embeddings maintain meaningful emotional semantic topology. Specifically, in *Brain27*, positive emotions are predominantly distributed in the bottom right, while negative emotions are distributed in the top left. In *Audio28*, *Bittersweet* is just located between *Sad* and other positive emotions. This visualization further demonstrates the necessity of modeling label dependencies.

**Augmented ERG Visualization.** In Figure 7(a), we provide the augmented ERG visulization on *Audio28* dataset. We utilize the oracle ERG, which is constructed using ground truth label statistics of all tasks, as the upper bound. We also compute the **P**earson's **C**orrelation **C**oefficients (PCCs) to measure the similarity between the augmented and oracle

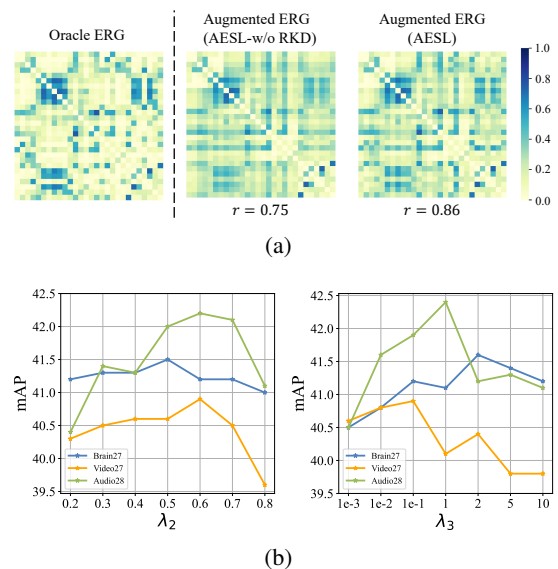

(a)

(b)

*Figure 7.* (a) Visualization of augmented emotional relation graph in *Audio28* dataset. (b) The performance of AESL measured by the last mAP changes as $\lambda_2$ and $\lambda_3$ vary.

ERG. Using the proposed method, inter- and intra-task label relationships are reconstructed well. Besides, by incorporating RKD from the affective dimension space, the augmented ERG is closer to the oracle ERG ($r = 0.86$ vs $r = 0.75$).

**Parameter Sensitivity.** Figure 7(b) gives an illustrative example of how the performance of AESL changes as the regulation parameters $\lambda_2$ and $\lambda_3$ vary on the B0-I9 and B0-I7 protocols of three datasets ($\lambda_1$ is fixed to 1). Here, when the value of one parameter varies, the other is fixed to a reasonable value. We find that too large value of $\lambda_2$ dramatically degrades the model performance due to the noise in affective ratings, while too small value will not play the role of alleviating the future-missing partial label problem. As for $\lambda_3$, too large value will cause the model to focus too much on graph reconstruction, while too small value will not be able to learn label embeddings well, which both lead to a decline in model performance.

## 5. Conclusion

In this paper, we have proposed a novel AESL framework for multi-label class incremental emotion decoding. In detail, we developed an augmented ERG generation method with label disambiguation for handling the past-missing partial label problem. Then, knowledge distillation from affective dimension space was introduced for alleviating the future-missing partial label problem. Besides, we constructed an emotional semantics learning module to learn indispensable label embeddings for subsequent semantic-specific feature extraction. Extensive experiments have illustrated the effectiveness of AESL.

## Acknowledgments

This work was supported in part by the National Key R&D Program of China 2022ZD0116500; in part by the National Natural Science Foundation of China under Grant 62206284; and in part by Beijing Natural Science Foundation under Grant L243016; and in part by the Beijing Nova Program under Grant 20230484460.

## Impact Statement

This research is motivated by the growing role of affective computing in shaping human-machine interactions. Advances in deep learning for emotion recognition and synthesis carry significant societal implications: Improved emotion-aware systems can enhance mental health support (e.g., AI-assisted therapy), education (e.g., adaptive learning systems), and human-computer collaboration (e.g., empathetic chatbots).

However, this technology presents inherent dual-use risks. While our models may help detect distress signals in clinical settings, the same techniques could be repurposed for unethical surveillance or manipulation (e.g., non-consensual emotional profiling). The accuracy-limitations of emotion models also raise concerns about cultural bias or misinterpretation of nuanced emotional states, potentially leading to harmful decisions in sensitive applications.

We argue that the benefits of responsible affective computing—particularly in healthcare and accessibility—outweigh the risks when accompanied by rigorous ethical safeguards. This work emphasizes the need for transparent model limitations, user consent protocols, and domain-specific deployment guidelines to mitigate misuse.

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

# A. The Algorithm of AESL

---

**Algorithm 1** Training procedure of AESL.

---

**Input:** Training sequence $\{\mathcal{D}^1, \cdots, \mathcal{D}^B\}$. Hyperparameters $\sigma$, $\beta$, $\lambda_1$, $\lambda_2$, $\lambda_3$. Affective dimension features $\{\boldsymbol{\tau}^1, \cdots, \boldsymbol{\tau}^B\}$.

**for** $b = 1 : B$ **do**

    **while** not converged **do**

        **for** $(\mathbf{x}^b, \mathbf{y}^b) \sim \mathcal{D}^b$ **do**

            **if** $b = 1$ **then**

                Compute $\mathbf{A}^b$ directy with label matrix $\mathbf{Y}^b$ using Eq.1.

            **else**

                Compute soft label matrix $\mathbf{S}$ with $\mathbf{s} = \psi(\mathbf{x}^b, \mathbf{A}^{b-1}; \Phi^{b-1})$ and set intial label confidence matrix $\mathbf{F}_0 = \mathbf{S}$.

                Compute the normalizing weight matrix $\hat{\mathbf{P}} = \mathbf{P}\mathbf{D}^{-1}$. $\mathbf{P}_{ij}$ is the similarity between two instances in $\mathcal{D}^b$.

                Implement label propagating to obatin refined soft label matrix $\hat{\mathbf{S}}$ with Eq.3.

                Compute $\mathbf{B}^b$ directy with label matrix $\mathbf{Y}^b$ using Eq.1.

                Compute $\mathbf{R}^b$ and $\mathbf{Q}^b$ with $\hat{\mathbf{S}}$ and $\mathbf{Y}^b$ using Eq.4 and 5, then obtain $\mathbf{A}^b = \begin{bmatrix} \mathbf{A}^{b-1} & \mathbf{R}^b \\ \mathbf{Q}^b & \mathbf{B}^b \end{bmatrix}$.

                {Get augmented ERG shown in Section 3.3.}

            **end if**

            Construct (augmented) ERG $\mathcal{G}^b$ with initial node features $\mathbf{H}_0^b$ and adjacency matrix $\mathbf{A}^b$.

            Implement message passing strategy using Eq.6 to obtain label semantic embeddings $\mathbf{E}^b$.

            {Implement label semantics learning shown in Section 3.4.}

            Compute importance vectors $\alpha$ using a fully-connected network followed by a sigmoid function.

            Compute semantic-specific features $\mathbf{o}$ with deep latent feature $\mathbf{z}$ and importance vectors $\alpha$ with Eq.8.

            Compute the label confidence scores $\mathbf{s}$ using Eq.9 to obtain the prediction for emotion classes $1, ..., \mathcal{C}^b$.

            {Implement semantic-guided feature decoupling to obtain semantic-specific features shown in Section 3.5.}

            Compute the representation similarity matrix $\mathbf{M}^b$, $\mathbf{M}^{b-1}$ and $\mathbf{M}^{\text{aff}}$ with Eq.10 and 11.

            {Implement relation-based knowledge distillation with affective dimension features shonw in Section 3.6.}

            Compute the final loss $\mathcal{L}$ with Eq.7, 12 and 13. Update AESL model by minimizing $\mathcal{L}$.

        **end for**

    **end while**

**end for**

---

# B. Details of Experiments

### B.1. Details of Datasets

*Brain27* is a visually evoked emotional brain activity dataset (Horikawa et al., 2020), which contains the blood-oxygen-level dependent (BOLD) responses of five subjects, who were shown 2196 video clips while functional Magnetic Resonance Imaging (fMRI) data were recorded. These data were collected using a 3T Siemens scanner with a multiband gradient Echo-Planar Imaging (EPI) sequence (TR, 2000ms; TE, 43ms; flip angle, 80 deg; FOV, 192×192 mm; voxel size, 2×2×2 mm; number of slices, 76; multiband factor, 4). The fMRI data was preprocessed and averaged with each video stimulus, which means the brain activity of one voxel is a scalar for a video stimulus.

*Video27* is an emotionally evocative visual dataset (Cowen & Keltner, 2017), which had been used to collect the brain activity in *Brain27*. This dataset contains 2196 videos whose durations ranged from 0.15s to 90s. Some video screenshots of *Video27* have been shown in Figure 1.

*Audio28* is an emotionally evocative auditory dataset (Cowen et al., 2020), which consists of 1841 music samples without lyrics. In these music clips, 1572 were selected from YouTube, 88 came from Howard Shore's *Lord of the Rings* soundtrack, and 181 came from Wagner's *Ring* cycle. These segments of music can convey strong feelings, whose durations ranged from 0.73s to 7.89s.

In terms of emotion category and affective dimension ratings, in *Brain27* and *Video27*, each instance was voted by multiple raters across 27 emotion categories and 14 affective dimensions. In *Audio28*, each music clip was judged by multiple rates across 28 emotion categories and 11 affective dimensions. Emotion category ratings range from 0 to 1 and we set threshold 0.1 for *Brain27* and *Video27* and 0.15 for *Audio28* to construct emotion label matrix. The average number of emotion labels for the former two datasets and *Audio28* is 4.64 and 5.27, respectively. Affective dimension ratings were rated by a 9-scale Likert scale, which are standardized before RKD in our experiments.

In the process of splitting emotion labels for incremental learning, we just follow the order of the alphabet without other interfere. The order of *Brain27* and *Video27* is *Admiration*, *Adoration*, *Aesthetic appreciation*, *Amusement*, *Anger*, *Anxiety*, *Awe*, *Awkwardness*, *Boredom*, *Calmness*, *Confusion*, *Craving*, *Disgust*,*Empathic pain*, *Entrancement*,*Excitement*, *Fear*, *Horror*, *Interest*, *Joy*, *Nostalgia*, *Relief*, *Romance*, *Sadness*, *Satisfaction*, *Sexual desire* and *Surprise*. The order of *Audio28* is *Amusing*, *Angry*, *Annoying*, *Anxious*, *Amazing*, *Beautiful*, *Bittersweet*, *Calm*, *Compassionate*, *Dreamy*, *Eerie*, *Energizing*, *Entrancing*, *Erotic*, *Euphoric*, *Exciting*, *Goose bumps*, *Indignant*, *Joyful*, *Nauseating*, *Painful*, *Proud*, *Romantic*, *Sad*, *Scary*, *Tender*, *Transcendent* and *Triumphant*. Definitions of the 27 and 28 emotion categories are detailed in Table 6.

Affective dimensions used in *Brain27* and *Video27* are *Approach*, *Arousal*, *Attention*, *Certainty*, *Commitment*, *Control*, *Dominance*, *Effort*, *Fairness*, *Identity*, *Obstruction*, *Safety*, *Upswing* and *Valence*. Affective dimensions used in *Audio28* are *Arousal*, *Attention*, *Certainty*, *Commitment*, *Dominance*, *Enjoyment*, *Familiarity*, *Identity*, *Obstruction*, *Safety* and *Valence*.

Table 7 shows the characteristics of the three datasets used in our experiments. Properties of each dataset are characterized by several statistics, including the number of training instances $|\mathcal{D}_{tr}|$, the number of test instances $|\mathcal{D}_{te}|$, the number of features $Dim(\mathcal{D})$, the threshold for constructing label matrix $Th(\mathcal{D})$, the number of possible class labels $L(\mathcal{D})$, the number of affective dimensions $Aff(\mathcal{D})$, the label cardinality (average number of labels per instance) $LCard(\mathcal{D})$, the label density (label cardinality over $L(\mathcal{D})$) $LDen(\mathcal{D})$, and the modality. For *Brain27*, we also exhibit the number of voxels $V(\mathcal{D})$ before ROI-pooling used in our experiments.

## B.2. Comparing Approaches

Details of these compared methods are as follows.

**EWC** (Kirkpatrick et al., 2017): A Single-Label Class Incremental Learning algorithm that reduces catastrophic forgetting by constraining important parameters uses the Fisher information matrix to compute the importance of parameters.

**LWF** (Lee et al., 2019): The first algorithm to apply knowledge distillation to the Single-Label Class Incremental Learning task uses the old model as a teacher and minimizes the KL divergence between the probability distributions of the outputs of the new and old models.

**ER** (Lee et al., 2019): A Single-Label Class Incremental Learning algorithm based on data replay, where the construction of a data buffer employs a random sampling strategy.

**RS** (Vitter, 1985): A Single-Label Class Incremental Learning algorithm based on data replay, where the construction of a data buffer utilizes a reservoir sampling strategy.

**AGCN** (Vitter, 1985): A Multi-Label Class Incremental Learning algorithm based on graph convolutional neural networks, where the graph adjacency matrix continuously expands as the tasks progress.

**PRS** (Kim et al., 2020): A Multi-Label Class Incremental Learning algorithm based on data replay, which improves upon the reservoir sampling strategy to ensure that the number of samples for each class in the data buffer is as balanced as possible.

**OCDM** (Liang & Li, 2022): A Multi-Label Class Incremental Learning algorithm based on data replay that defines the construction and updating of the data buffer as an optimization problem to be solved.

**KRT-R** (Dong et al., 2023): A Multi-Label Class Incremental Learning algorithm based on Knowledge Restore and Transfer (KRT) framework.

## B.3. Feature Extraction

In *Brain27* dataset, we extract a 2880-dimensional feature vector with ROI-pooling (see Figure 8). In *Video27*, visual object features have been extracted with a pre-trained VGG19 model (Simonyan & Zisserman, 2014) for one frame and averaged across all frames to construct 1000-dimensional features. In *Audio28* dataset, we compute **M**el-**f**requency **c**epstral

*Table 6.* Definitions of emotion categories for *Brain27*, *Video28*, and *Audio28*.

| | Name | Definition |
|---|---|---|
| **27 emotion categories** | Admiration | A feeling of deep respect and appreciation for someone's qualities or achievements. |
| | Adoration | A profound sense of love, devotion, or reverence for someone or something cherished. |
| | Aesthetic appreciation | The pleasure and admiration felt when encountering beauty or artistic excellence. |
| | Amusement | A lighthearted and joyful response to something funny or entertaining. |
| | Anger | A strong emotional reaction to perceived harm, injustice, or frustration. |
| | Anxiety | A tense, uneasy feeling often associated with fear or worry about uncertain outcomes. |
| | Awe | A profound emotional response to something vast, grand, or beyond ordinary comprehension. |
| | Awkwardness | A sense of discomfort or embarrassment in socially clumsy or uncertain situations. |
| | Boredom | A state of weariness or dissatisfaction caused by a lack of interest or engagement. |
| | Calmness | A serene, peaceful state of mind free from stress or agitation. |
| | Confusion | A feeling of bewilderment or uncertainty when faced with something unclear or unexpected. |
| | Craving | An intense desire or longing for something specific, often food or experiences. |
| | Disgust | A strong feeling of aversion or revulsion, often triggered by something offensive or unpleasant. |
| | Empathic pain | An emotional resonance with another's suffering, leading to shared feelings of distress. |
| | Entrancement | A captivating, hypnotic feeling that draws one into a deeply absorbing experience. |
| | Excitement | A heightened state of anticipation or enthusiasm about something exhilarating or enjoyable. |
| | Fear | A powerful emotion in response to perceived danger or threat, prompting a fight-or-flight reaction. |
| | Horror | An intense fear mixed with shock or revulsion, often caused by something terrifying or gruesome. |
| | Interest | A sense of curiosity and attention sparked by something engaging or thought-provoking. |
| | Joy | A profound and uplifting feeling of happiness or delight. |
| | Nostalgia | A bittersweet longing for past experiences, often accompanied by fond memories. |
| | Relief | A lightened and eased feeling after the alleviation of stress, pain, or worry. |
| | Romance | A tender and affectionate emotion linked to love and intimate connection. |
| | Sadness | A heavy, sorrowful feeling typically associated with loss, disappointment, or empathy. |
| | Satisfaction | A contented sense of fulfillment after achieving a goal or desire. |
| | Sexual desire | A deep, physical and emotional yearning for intimacy and connection. |
| | Surprise | A sudden and unexpected emotion elicited by an unforeseen event or realization. |
| **28 emotion categories** | Amusing | This emotion brings a light-hearted, fun feeling, often sparking smiles and laughter. |
| | Angry | A powerful emotion, usually triggered by frustration or injustice, that drives a strong sense of displeasure. |
| | Annoying | A mild irritation or frustration caused by something persistently bothersome or inconvenient. |
| | Anxious | A feeling of worry, nervousness, or unease about an uncertain outcome or future event. |
| | Amazing | A sense of wonder and admiration often evoked by something remarkable or extraordinary. |
| | Beautiful | An emotion tied to the appreciation of visual, auditory, or conceptual harmony and appeal. |
| | Bittersweet | A mixed emotion of happiness and sadness, usually arising from nostalgia or a cherished memory. |
| | Calm | A soothing, peaceful feeling of tranquility and lack of disturbance. |
| | Compassionate | A warm, empathetic response that involves caring deeply for someone else's suffering. |
| | Dreamy | A relaxed, whimsical feeling, often associated with a sense of escapism or fantasy. |
| | Eerie | A feeling of mystery tinged with unease, often evoked by something strange or uncanny. |
| | Energizing | A rush of motivation and vitality that makes one feel alert and ready for action. |
| | Entrancing | A mesmerizing, absorbing emotion that captivates one's full attention. |
| | Erotic | A deeply intimate, sensual feeling characterized by physical and emotional desire. |
| | Euphoric | An intense, exhilarating joy that feels almost transcendent or surreal. |
| | Exciting | A feeling of lively anticipation and enthusiasm about something anticipated or unfolding. |
| | Goose bumps | A physical reaction to intense emotions, often linked to awe, fear, or admiration. |
| | Indignant | A righteous anger or resentment, typically provoked by perceived unfair treatment. |
| | Joyful | A pure, light-hearted happiness that uplifts and brightens one's mood. |
| | Nauseating | A strong feeling of physical discomfort, often coupled with disgust or revulsion. |
| | Painful | An intense, often distressing sensation caused by physical or emotional suffering. |
| | Proud | A positive feeling of satisfaction and fulfillment, often in recognition of an achievement. |
| | Romantic | A tender, affectionate feeling centered around love and connection. |
| | Sad | A heavy, sorrowful feeling often caused by loss, disappointment, or empathy. |
| | Scary | A strong, unsettling sense of fear triggered by a perceived threat or danger. |
| | Tender | A gentle, warm-hearted feeling of affection and care. |
| | Transcendent | An emotion that goes beyond ordinary experience, often bringing a sense of awe or enlightenment. |
| | Triumphant | A victorious, celebratory emotion after overcoming a challenge or achieving success. |

*Table 7.* The characteristics of the experimental datasets.

| Dataset | $|\mathcal{D}_{tr}|$ | $|\mathcal{D}_{te}|$ | $V(\mathcal{D})$ | $Dim(\mathcal{D})$ | $Th(\mathcal{D})$ | $L(\mathcal{D})$ | $Aff(\mathcal{D})$ | $LCard(\mathcal{D})$ | $LDen(\mathcal{D})$ | Modality |
|---|---|---|---|---|---|---|---|---|---|---|
| *Brain27*(Subject1) | 1800 | 396 | 120930 | 2880 | 0.10 | 27 | 14 | 4.64 | 0.17 | fMRI |
| *Brain27*(Subject2) | 1800 | 396 | 116260 | 2880 | 0.10 | 27 | 14 | 4.64 | 0.17 | fMRI |
| *Brain27*(Subject3) | 1800 | 396 | 102941 | 2880 | 0.10 | 27 | 14 | 4.64 | 0.17 | fMRI |
| *Brain27*(Subject4) | 1800 | 396 | 118533 | 2880 | 0.10 | 27 | 14 | 4.64 | 0.17 | fMRI |
| *Brain27*(Subject5) | 1800 | 396 | 116699 | 2880 | 0.10 | 27 | 14 | 4.64 | 0.17 | fMRI |
| *Video27* | 1800 | 396 | - | 1000 | 0.10 | 27 | 11 | 4.64 | 0.17 | Video |
| *Audio28* | 1500 | 341 | - | 512 | 0.15 | 28 | 11 | 5.27 | 0.19 | Audio |

*Figure 8.* A schematic diagram of ROI pooling. Each orange small cube represents a voxel. Using the brain voxels signal directly for voxel-wise decoding will introduce lots of noise and easily cause overfitting. Therefore, we first use the HCP360 template (Glasser et al., 2016) to divide the whole brain into multiple brain regions (ROIs) that include 360 cortical regions defined by a parcellation provided from the Human Connectome Project. In order to further extract the features of each ROI, we place the voxels of each ROI in a 3-D volume according to its coordinates, then split the volume evenly into 8 sub-volumes and calculate the average brain activity of voxels in each sub-volume as the feature of this sub-volume as illustrated in Figure 8. In other words, for each ROI we get 8-dimensional features. Then we concatenate the features of 360 ROIs in each hemisphere to get 2880-dimensional features.

coefficients (MFCC) for each audio fragment. All MFCC fragments from the same audio are then input into a pre-trained ResNet-18 (He et al., 2016) model and averaged across all fragments to obtain 512-dimensional features.

### B.4. Hyperparameters Settings

In our experiments, the balancing parameter $\beta$ is set to 0.95 in Eq.3. We set $\lambda_1$ to 1 in Eq.15. Besides, $\lambda_2$ is searched in $\{0.2, 0.3, 0.4, 0.5, 0.6, 0.7, 0.8\}$ and $\lambda_3$ is searched in $\{0.001, 0.01, 0.1, 1, 2, 5, 10\}$. The dimensionality of deep latent representations $\mathbf{z}$ is set to 64 in three datasets. We train the model using the Adam optimizer with $\{\beta_1, \beta_2\} = \{0.9, 0.9999\}$. We set weight decay to 0.005 and learning rate to $10^{-4}$ for *Brain27* and *Video27*, and weight decay of 0 and learning rate to $10^{-3}$ for *Audio28*. We conducted all the experiments on one NVIDIA TITAN GPU.

## C. More Comparative Results

Tables 8, 9, 10 and 11 show the results on subject 2, subject 3, subject4 and subject 5 in *Brain27* dataset. Figure 9 exhibits the comparison curves of AESL and comparing methods for these subjects in *Brain27* dataset. We observe that similar conclusions can be drawn as mentioned in Section 4.2.

## D. Limitations

At the application level, in the affective HCI tasks in real-life scenarios, in addition to learning new emotion categories, we also need to adapt to new subjects. Therefore, it is necessary to further consider the continuous learning of emotions from different subjects, which can be regarded as domain incremental learning. At the experimental level, the impact of the order of learning emotion categories and the number of emotion categories in each task on the experimental results needs to be further explored.

*Table 8.* Class incremental results on subject 2 of *Brain27* dataset. AGCN, PRS and OCDM are MLCIL algorithms among these compared methods.

| Method | Brain27 B0-I9 | | | | Brain27 B0-I3 | | | | Brain27 B15-I3 | | | | Brain27 B15-I2 | | | |
|---|---|---|---|---|---|---|---|---|---|---|---|---|---|---|---|---|
| | Avg. Acc | Last Acc | | | Avg. Acc | Last Acc | | | Avg. Acc | Last Acc | | | Avg. Acc | Last Acc | | |
| | mAP | maF1 | miF1 | mAP | mAP | maF1 | miF1 | mAP | mAP | maF1 | miF1 | mAP | mAP | maF1 | miF1 | mAP |
| Upper-bound | - | 34.0 | 44.3 | 42.3 | - | 34.0 | 44.3 | 42.3 | - | 34.0 | 44.3 | 42.3 | - | 34.0 | 44.3 | 42.3 |
| Finetune | 34.9 | 8.2 | 18.5 | 24.8 | 31.0 | 5.0 | 14.0 | 21.0 | 25.8 | 5.1 | 14.4 | 20.0 | 23.4 | 3.7 | 13.3 | 18.7 |
| EWC | 34.3 | 7.8 | 18.1 | 25.2 | 31.2 | 5.0 | 13.7 | 21.7 | 26.4 | 5.0 | 13.8 | 20.1 | 24.1 | 3.7 | 13.3 | 19.4 |
| LwF | 38.9 | 11.4 | 28.6 | 29.6 | 37.0 | 19.4 | 37.0 | 25.9 | 31.8 | 15.7 | 32.5 | 24.4 | 29.4 | 15.4 | 32.6 | 21.9 |
| ER | 40.4 | 8.5 | 13.1 | 34.9 | 39.3 | 3.8 | 4.8 | 20.8 | 37.3 | 6.9 | 11.6 | 34.1 | 36.3 | 7.4 | 11.0 | 32.7 |
| RS | 41.0 | 7.6 | 10.8 | 35.9 | 40.4 | 3.7 | 5.9 | 28.9 | 36.4 | 6.2 | 10.3 | 31.4 | 35.8 | 4.3 | 7.8 | 31.0 |
| AGCN | 43.4 | 28.8 | 43.1 | 39.8 | 43.0 | 32.4 | 43.6 | 33.7 | 39.0 | 24.8 | 39.5 | 35.2 | 37.3 | 24.0 | 35.5 | 32.1 |
| PRS | 40.4 | 7.9 | 14.4 | 34.4 | 40.9 | 3.6 | 7.8 | 20.2 | 37.8 | 8.4 | 12.0 | 34.1 | 37.2 | 8.5 | 11.8 | 32.5 |
| OCDM | 40.0 | 8.4 | 12.9 | 33.4 | 40.5 | 4.5 | 8.1 | 28.2 | 36.1 | 5.9 | 9.2 | 30.9 | 34.7 | 5.1 | 8.6 | 29.4 |
| **AESL** | **45.4** | **29.8** | **43.2** | **41.4** | **45.5** | **32.5** | **44.2** | **35.3** | **40.2** | **25.5** | **40.1** | **36.4** | **39.3** | **24.8** | **36.4** | **35.0** |

*Table 9.* Class incremental results on subject 3 of *Brain27* dataset. AGCN, PRS and OCDM are MLCIL algorithms among these compared methods.

| Method | Brain27 B0-I9 | | | | Brain27 B0-I3 | | | | Brain27 B15-I3 | | | | Brain27 B15-I2 | | | |
|---|---|---|---|---|---|---|---|---|---|---|---|---|---|---|---|---|
| | Avg. Acc | Last Acc | | | Avg. Acc | Last Acc | | | Avg. Acc | Last Acc | | | Avg. Acc | Last Acc | | |
| | mAP | maF1 | miF1 | mAP | mAP | maF1 | miF1 | mAP | mAP | maF1 | miF1 | mAP | mAP | maF1 | miF1 | mAP |
| Upper-bound | - | 33.6 | 44.2 | 42.0 | - | 33.6 | 44.2 | 42.0 | - | 33.6 | 44.2 | 42.0 | - | 33.6 | 44.2 | 42.0 |
| Finetune | 33.0 | 7.7 | 17.3 | 25.8 | 29.4 | 5.0 | 14.0 | 20.5 | 25.0 | 5.3 | 14.0 | 20.0 | 23.2 | 4.1 | 13.8 | 19.2 |
| EWC | 32.4 | 7.7 | 16.7 | 25.3 | 29.6 | 5.2 | 13.8 | 19.8 | 25.5 | 5.3 | 14.0 | 20.0 | 24.0 | 4.0 | 13.6 | 19.6 |
| LwF | 36.4 | 11.7 | 28.7 | 30.3 | 34.7 | 21.7 | 38.3 | 23.5 | 30.3 | 15.4 | 31.5 | 23.6 | 28.2 | 15.6 | 31.8 | 21.3 |
| ER | 37.1 | 7.0 | 10.2 | 33.5 | 36.7 | 2.4 | 2.4 | 30.5 | 34.6 | 5.0 | 8.6 | 31.9 | 35.4 | 6.8 | 10.9 | 32.9 |
| RS | 38.1 | 6.5 | 9.2 | 35.8 | 37.7 | 2.9 | 5.2 | 29.0 | 34.1 | 5.0 | 7.7 | 30.3 | 33.8 | 3.6 | 6.8 | 30.0 |
| AGCN | 40.2 | 27.6 | 42.0 | 38.2 | 40.1 | 32.0 | 42.9 | 32.7 | 37.0 | 27.0 | 37.1 | 33.6 | 34.9 | 24.1 | 36.6 | 30.2 |
| PRS | 38.0 | 7.6 | 12.0 | 34.3 | 38.8 | 3.6 | 6.0 | 29.5 | 35.5 | 5.5 | 7.7 | 32.7 | 36.0 | 4.5 | 7.3 | 33.1 |
| OCDM | 37.9 | 8.4 | 11.7 | 35.0 | 38.0 | 5.4 | 8.2 | 30.1 | 34.4 | 4.0 | 6.0 | 30.3 | 33.2 | 3.1 | 5.3 | 28.5 |
| **AESL** | **42.6** | **28.5** | **44.0** | **40.8** | **42.3** | **33.4** | **43.2** | **33.4** | **38.8** | **27.2** | **38.9** | **35.4** | **37.9** | **25.5** | **37.7** | **34.1** |

*Table 10.* Class incremental results on subject 4 of *Brain27* dataset. AGCN, PRS and OCDM are MLCIL algorithms among these compared methods.

| Method | Brain27 B0-I9 | | | | Brain27 B0-I3 | | | | Brain27 B15-I3 | | | | Brain27 B15-I2 | | | |
|---|---|---|---|---|---|---|---|---|---|---|---|---|---|---|---|---|
| | Avg. Acc | Last Acc | | | Avg. Acc | Last Acc | | | Avg. Acc | Last Acc | | | Avg. Acc | Last Acc | | |
| | mAP | maF1 | miF1 | mAP | mAP | maF1 | miF1 | mAP | mAP | maF1 | miF1 | mAP | mAP | maF1 | miF1 | mAP |
| Upper-bound | - | 38.3 | 48.6 | 45.1 | - | 38.3 | 48.6 | 45.1 | - | 38.3 | 48.6 | 45.1 | - | 38.3 | 48.6 | 45.1 |
| Finetune | 36.0 | 7.9 | 19.0 | 25.2 | 30.9 | 4.9 | 13.7 | 21.1 | 26.4 | 4.6 | 13.4 | 20.3 | 23.9 | 3.7 | 13.2 | 18.1 |
| EWC | 35.0 | 7.9 | 18.5 | 25.3 | 31.7 | 5.1 | 13.8 | 22.0 | 27.2 | 4.8 | 13.7 | 20.4 | 25.1 | 3.7 | 13.2 | 20.1 |
| LwF | 39.6 | 15.1 | 33.9 | 30.8 | 37.5 | 20.5 | 37.5 | 23.1 | 33.3 | 17.2 | 32.8 | 25.2 | 30.9 | 15.8 | 33.5 | 22.7 |
| ER | 42.0 | 10.1 | 15.6 | 37.7 | 41.8 | 3.9 | 4.8 | 34.8 | 40.1 | 9.3 | 11.3 | 37.0 | 38.8 | 9.7 | 12.4 | 36.4 |
| RS | 42.6 | 9.0 | 14.2 | 38.1 | 43.5 | 4.3 | 7.1 | 34.4 | 38.8 | 7.5 | 10.8 | 34.6 | 37.8 | 5.4 | 10.6 | 33.2 |
| AGCN | 43.7 | 31.8 | 46.4 | 40.9 | 44.1 | 36.1 | 44.5 | 36.3 | 41.0 | 30.3 | 43.7 | 38.6 | 39.3 | 26.6 | 37.1 | 35.0 |
| PRS | 43.5 | 10.7 | 15.9 | 39.9 | 44.2 | 4.0 | 5.6 | 32.8 | 40.2 | 8.5 | 11.1 | 36.1 | 39.3 | 9.1 | 12.2 | 35.6 |
| OCDM | 43.4 | 11.4 | 15.1 | 39.8 | 43.0 | 4.9 | 7.8 | 31.7 | 38.2 | 6.0 | 5.9 | 33.4 | 38.4 | 5.2 | 9.4 | 33.4 |
| **AESL** | **45.7** | **36.4** | **47.3** | **44.0** | **47.1** | **37.2** | **46.4** | **38.7** | **43.4** | **31.3** | **44.4** | **39.9** | **42.2** | **29.0** | **40.4** | **37.7** |

*Table 11.* Class incremental results on subject 5 of *Brain27* dataset. AGCN, PRS, OCDM are MLCIL algorithm in these compared methods.

| Method | Brain27 B0-I9 | | | | Brain27 B0-I3 | | | | Brain27 B15-I3 | | | | Brain27 B15-I2 | | | |
|---|---|---|---|---|---|---|---|---|---|---|---|---|---|---|---|---|
| | Avg. Acc | Last Acc | | | Avg. Acc | Last Acc | | | Avg. Acc | Last Acc | | | Avg. Acc | Last Acc | | |
| | mAP | maF1 | miF1 | mAP | mAP | maF1 | miF1 | mAP | mAP | maF1 | miF1 | mAP | mAP | maF1 | miF1 | mAP |
| Upper-bound | - | 36.6 | 46.8 | 45.0 | - | 36.6 | 46.8 | 45.0 | - | 36.6 | 46.8 | 45.0 | - | 36.6 | 46.8 | 45.0 |
| Finetune | 34.7 | 7.3 | 17.8 | 25.4 | 31.1 | 5.4 | 14.3 | 21.7 | 25.4 | 5.2 | 13.9 | 18.9 | 23.4 | 3.9 | 13.2 | 19.2 |
| EWC | 34.1 | 6.7 | 15.6 | 25.1 | 31.0 | 4.9 | 13.5 | 21.9 | 26.3 | 5.0 | 13.7 | 20.3 | 24.3 | 4.0 | 13.5 | 20.2 |
| LwF | 37.6 | 12.2 | 30.3 | 28.7 | 35.5 | 21.0 | 37.9 | 25.8 | 31.2 | 15.8 | 31.7 | 24.6 | 29.4 | 14.3 | 29.0 | 22.4 |
| ER | 40.7 | 8.8 | 13.9 | 35.8 | 41.7 | 4.5 | 7.6 | 34.0 | 38.1 | 6.9 | 8.9 | 36.4 | 36.3 | 6.9 | 9.8 | 32.9 |
| RS | 41.4 | 7.6 | 11.3 | 36.9 | 41.5 | 4.4 | 7.6 | 31.7 | 37.1 | 5.7 | 8.6 | 33.4 | 37.3 | 5.4 | 9.0 | 33.0 |
| AGCN | 43.8 | 29.5 | 44.0 | 41.4 | 42.9 | 32.5 | 42.5 | 34.1 | 38.4 | 28.7 | 41.5 | 35.0 | 37.3 | 24.6 | 37.6 | 33.1 |
| PRS | 41.2 | 8.9 | 16.4 | 37.0 | 42.4 | 5.3 | 8.3 | 32.0 | 37.8 | 6.7 | 9.0 | 34.1 | 38.4 | 6.5 | 9.4 | 34.9 |
| OCDM | 41.1 | 8.3 | 14.0 | 35.5 | 41.3 | 5.4 | 8.0 | 29.7 | 36.8 | 4.4 | 7.9 | 31.5 | 34.8 | 4.6 | 8.0 | 30.1 |
| **AESL** | **46.0** | **32.9** | **45.8** | **44.1** | **45.5** | **35.3** | **45.6** | **35.4** | **40.4** | **29.5** | **44.5** | **37.5** | **39.9** | **29.3** | **40.8** | **36.2** |

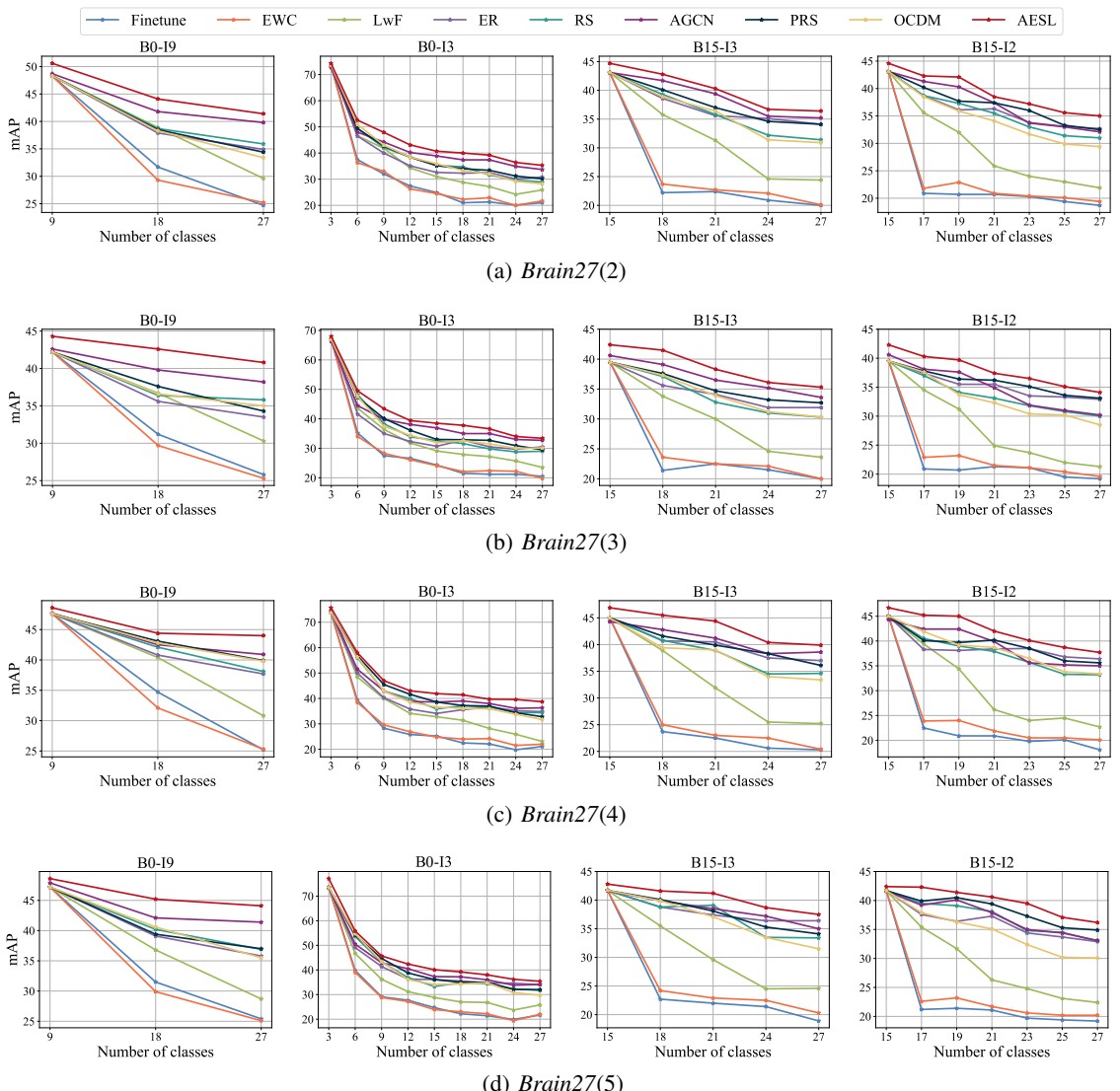

*Figure 9.* Comparison results (mAP) on three datasets used in our experiment under different protocols against compared CIL methods.

