# OpenReview forum: "EmoGrowth: Incremental Multi-label Emotion Decoding with Augmented Emotional Relation Graph"
_ICML.cc/2025/Conference — ICML 2025 poster_

### Official Review · Reviewer_W2Hb · 2025-03-12

**Overall Recommendation:** 3

**Summary:**

The author proposes an Augmented Emotional Semantic Learning (AESL) framework, which incorporates an Emotion Relation Graph (ERG) to enhance emotion classification. To address the issue of missing partial labels in past data, a reliable soft label generation method is introduced. Additionally, a Relation-based Knowledge Distillation (RKD) approach is proposed to mitigate the impact of missing labels in future data. Furthermore, this work pioneers the application of Multi-Label Class-Incremental Learning (MLCIL) to real-world emotion classification tasks. The effectiveness of the proposed framework is demonstrated through various incremental learning protocols on three different datasets.

## update after rebuttal
I would like to keep my rating.

**Claims And Evidence:**

The related work and ablation experiments provide theoretical support for the various components of this complex model and demonstrate their effectiveness in practice. However, the overall presentation lacks clarity and rigor.

**Essential References Not Discussed:**

No

**Experimental Designs Or Analyses:**

The compared works have not publicly reported experiments on these three datasets. Therefore, it is unclear whether the experimental results under the customized protocol adhere to the optimal settings of the respective baseline methods.

**Methods And Evaluation Criteria:**

Yes. The proposed method is relatively novel, and the experiments are fairly comprehensive.

**Other Comments Or Suggestions:**

No

**Other Strengths And Weaknesses:**

Strengths:
1. The paper introduces a novel and dynamic Emotion Relation Graph (ERG) that utilizes cross-attention to capture correlations between new and old classes. A label propagation algorithm is used to iteratively refine soft labels, while a Graph Autoencoder (GAE) learns semantic embeddings of emotion labels. Furthermore, high-dimensional alignment is performed within the Valence-Arousal feature space. These combined methods effectively mitigate the problem of catastrophic forgetting.
2. The proposed framework is thoroughly evaluated on three datasets, with extensive ablation experiments and visualization provided.

Weaknesses:
1. The framework presented is relatively complex, with significant interdependence between its modules. However, the explanations of the associated figures are somewhat vague, and there is insufficient discussion of their relevance. Is the ERG specifically constructed for this task accurate?
2. The overall content presentation and sequence of the paper lack clarity. For example, the definition of ERG should be introduced prior to Section 3.3.
3. The compared works have not publicly conducted experiments on the three datasets used in this study. Therefore, it is unclear whether the experimental results align with their optimal configurations.

**Questions For Authors:**

1. I would like to understand the practical significance of the Multi-Label Class Incremental Emotion Decoding field. Many existing emotion-related methods are capable of integrating different tasks into a single input for large models or other frameworks.
2. Please refer to the weaknesses above.

**Relation To Broader Scientific Literature:**

Research in the field of Multi-Label Class Incremental Emotion Decoding is relatively scarce, and this paper appears to be among the early studies to focus on this area.

**Theoretical Claims:**

The content description and organization of the paper are somewhat disordered. For example, the definition of ERG in Section 3.4 should be introduced before Section 3.3.

---

> ### Author Rebuttal · Authors · 2025-03-31
>
> **1.I would like to understand the practical significance of the Multi-Label Class Incremental Emotion Decoding field. Many existing emotion-related methods are capable of integrating different tasks into a single input for large models or other frameworks.**
>
>  (1) In real world Brain-Computer Interfaces, practical applications often require dynamic adaptation to new emotion categories over time. For example:
>
> •  In clinical BCIs for depression monitoring, initial models may focus on basic emotions (e.g., happy/sad), but later need to incorporate nuanced states (e.g., anhedonia, emotional blunting) as therapy progresses.
>
> •  Consumer-grade BCIs (e.g., gaming or education) may start with coarse emotion labels but require incremental addition of context-specific labels (e.g., "frustration during learning" or "flow state").
>
> (2) Although large language models (LLMs) demonstrate strong performance in multi-task integration, many studies have shown that their continual learning capabilities remain limited and often suffer from catastrophic forgetting[1][2]. Therefore, investigating incremental learning remains crucial, particularly for specialized models in affective computing.
>
> **2.The explanations of the associated figures are somewhat vague, and there is insufficient discussion of their relevance. Is the ERG specifically constructed for this task accurate?**
>
> We sincerely apologize for any confusion caused. To clarify:
>
> •  Figure 2(a) presents the overall model architecture
>
> •  Figure 2(b) provides detailed illustration of the AEG-D component marked in Figure 2(a)
>
> •  Figure 3 specifically shows the knowledge distillation scheme introduced to mitigate catastrophic forgetting caused by future label absence, which contributes additional loss functions for better training of the Figure 2(a) framework.
>
> We will enhance the explanation of these relationships in the final version.
>
> For ERG, as evidenced in Figure 5, the learned emotion embeddings maintain meaningful topological structures in the semantic space, which illustrates the accuracy of the ERG.
>
> **3.The overall content presentation and sequence of the paper lack clarity. For example, the definition of ERG should be introduced prior to Section 3.3.**
>
> We appreciate the reviewer's suggestion and will adjust the manuscript to present ERG prior to Section 3.3 for better logical flow in the final version.
>
> **4.The compared works have not publicly conducted experiments on the three datasets used in this study. Therefore, it is unclear whether the experimental results align with their optimal configurations.**
>
> All compared methods maintain their original model architectures as reported in their respective papers. For feature extraction, a critical component in emotion decoding, we adopted the most widely-used approaches for these three datasets, as established in prior literature[3][4]. For parameter optimization, we reserved a separate validation set and carefully selected optimal configurations for fair comparison.
>
> **References:**
>
> [1] Chen W, Zhou Y, Du N, et al. Lifelong language pretraining with distribution-specialized experts[C]//International Conference on Machine Learning. PMLR, 2023: 5383-5395.
>
> [2] Hu H, Sener O, Sha F, et al. Drinking from a firehose: Continual learning with web-scale natural language[J]. IEEE Transactions on Pattern Analysis and Machine Intelligence, 2022, 45(5): 5684-5696.
>
> [3] Fu K, Du C, Wang S, et al. Multi-view multi-label fine-grained emotion decoding from human brain activity[J]. IEEE Transactions on Neural Networks and Learning Systems, 2022.
>
> [4] Horikawa T, Cowen A S, Keltner D, et al. The neural representation of visually evoked emotion is high-dimensional, categorical, and distributed across transmodal brain regions[J]. Iscience, 2020, 23(5).
>
> [5] Cowen A S, Fang X, Sauter D, et al. What music makes us feel: At least 13 dimensions organize subjective experiences associated with music across different cultures[J]. Proceedings of the National Academy of Sciences, 2020, 117(4): 1924-1934.

---

### Official Review · Reviewer_dimC · 2025-03-12

**Overall Recommendation:** 3

**Summary:**

This paper introduces multi-label fine-grained class incremental emotion decoding, which aims to develop models capable of incrementally learning new emotion categories while maintaining the ability to recognize multiple concurrent emotions. It proposes an Augmented Emotional Semantics Learning (AESL) framework to address two critical challenges: past- and future-missing partial label problems. AESL incorporates an augmented Emotional Relation Graph (ERG) for reliable soft label generation and affective dimension-based knowledge distillation for future-aware feature learning. Experiments demonstrate the effectiveness of the proposed method. The main contributions of this paper are:

- It introduces the multi-label class incremental emotion decoding.
- It develops an innovative augmented emotional semantics learning framework.

## update after rebuttal
The authors' rebuttal addresses most of my concerns. I would like to keep my rating and weakly support the acceptance of the paper.

**Claims And Evidence:**

The claims are supported by evidence. The proposed method's superiority is evident in its consistent outperformance of existing methods.

**Essential References Not Discussed:**

There are no essential references that are missing from the paper.

**Experimental Designs Or Analyses:**

The experimental designs and analyses are overall reasonable. However, the parameter sensitivity analyses are missing. It is not clear how the regularization term $\lambda_1$ affect the final performance.

**Methods And Evaluation Criteria:**

The proposed methods and evaluation criteria make sense for the studied problem. The proposed AESL is designed to handle the specific challenges of multi-label class incremental emotion decoding, and the evaluation metrics are appropriate for assessing the performance of the proposed method.

**Other Comments Or Suggestions:**

Providing more details on computational requirements and efficiency can further strengthen the paper. Moreover, discussing the limitations of the proposed method and potential future research directions can better facilitate the reader's understanding.

**Other Strengths And Weaknesses:**

### Strengths

- The problem studied in this paper is interesting.

- This paper is well written and in good sharp, which is easy to follow.

- The experimental results are somehow promising.



### Weaknesses

- This paper introduces a novel research problem, multi-label class incremental emotion decoding. However, the unique challenges of this problem compared to similar problems are not explicitly pointed out, and the significance of this problem also needs to be further elaborated.
- From Table 4, one can observe that the performance of "w/o ESL&LD+AD" is very close to AESL and better than other cases. The authors should analyze the reason for this phenomenon.
- It is not clear how the hyperparameter $\lambda_1$ affects the final performance.

**Questions For Authors:**

1. What are the unique challenges of multi-label class incremental emotion decoding compared to similar problems?
2. From Table 4, one can observe that the performance of "w/o ESL&LD+AD" is very close to AESL and better than other cases. The authors should analyze the reason for this phenomenon.
3. How do the hyperparameter $\lambda_1$ affect the final performance?

**Relation To Broader Scientific Literature:**

The paper's contributions are relevant to the broader scientific literature in affective computing and machine learning. It builds on existing research in class incremental learning and multi-label classification, advancing the field by addressing specific challenges in emotion decoding.

**Theoretical Claims:**

Theoretical statements are not involved in this paper.

---

> ### Author Rebuttal · Authors · 2025-03-31
>
> **1.What are the unique challenges of multi-label class incremental emotion decoding compared to similar problems?**
>
> (1) Unlike traditional single-label class incremental learning, multi-label class incremental faces unique challenges in addressing catastrophic forgetting from past- and future-missing partial label problems. Specifically, while a single sample may correspond to multiple emotion categories, only a specific subset is available during each task phase. For past labels, if the model fails to reconstruct them, all previous emotion categories would be incorrectly treated as negative samples, significantly degrading model performance. For future labels, the model requires the incorporation of domain knowledge to develop certain reasoning capabilities about unknown categories.
> (2) Multi-label class-incremental emotion decoding deals with highly fine-grained emotional categories. In typical scenarios, human emotional variations are remarkably subtle, causing samples from different categories to be distributed in closer within the feature space. Consequently, decoding such fine-grained emotional categories presents significant challenges.
>
> **2.From Table 4, one can observe that the performance of "w/o ESL&LD+AD" is very close to AESL and better than other cases. The authors should analyze the reason for this phenomenon.**
>
> “w/o ESL&LD+AD” means that directly integrate the sample-wise affective dimension into category-wise emotion embeddings. In this setup, we aim to explore the rationale of incorporating domain knowledge on affective dimension to mitigate the issue of future missing partial label problem. In fact, the affect dimension space can represent an arbitrary number of emotion categories, so directly using affective dimension features for class-incremental emotion decoding would yield excellent results. However, in practical applications, this approach requires pre-establishing a mapping between emotion categories and affective dimensions, rather than constituting an end-to-end model.
>
> **3.How do the hyperparameter λ1 affect the final performance?**
>
> In fact, in the loss function shown in Equation (15), the relative magnitudes of λ₁, λ₂, and λ₃ are quite important. In the Brain27 dataset with B0-I9 setting, we fixed λ₂ = 0.5 and λ₃ = 2, and observed the impact of varying λ₁ on model performance as follows:
>
> | $\lambda_1$ | 0.2   | 0.3   | 0.4  | 0.5  | 0.6  |  0.7  |  0.8  |
> |-----------------|---------|---------|--------|-------|--------|----------|---------|
> | **mAP (%)** | 41.4 | 41.9  |42.7 | 42.9  | 43.1 |  42.6 |  42.0  |
>
> We can observe that an excessively small λ₁ leads the model to distill more information from the affective dimension space. Considering the heterogeneity of the feature space, this makes the model harder to train and results in lower accuracy. On the other hand, an overly large λ₁ makes the model more susceptible to the impact of future-missing partial label problem, thereby degrading its performance.

---

### Official Review · Reviewer_DLjo · 2025-03-13

**Overall Recommendation:** 2

**Summary:**

The paper introduces multi-label, fine-grained class incremental emotion decoding AESL to adapt to the scenarios where novel emotion categories continuously emerge. To solve the critical past-missing partial label problem, AESL introduces an augmented Emotional Relation Graph (ERG) module, using graph-based label disambiguation to generate reliable soft labels. AESL enhances ERG by integrating historical ERG with new data, preserving emotional label correlations. Moreover, a relation-based knowledge distillation framework is proposed to align model features with the affective dimension space. The emotional semantics learning module utilizes ERG to design a graph autoencoder and learns emotion embeddings to facilitate semantic-specific feature decoupling. Comprehensive evaluations across three datasets (Brain27, Video27, and Audio28) prove the effectiveness of the proposed AESL.

## update after rebuttal
The rebuttal addresses several concerns, including category order on the original tasks, teacher models, and CSC comparison. However, it falls short in providing a discussion/experiments on varying granularity level of tasks (2-cls sentiment vs 7-cls basic emotion vs 28-cls emotion), a motivation with evidence to apply class incremental learning for emotion analysis, and a literature review & SOTA comparison with recent works. The aforementioned lack limits the robustness of their claims. The explanation of motivation also lacks depth, especially regarding its significance and characteristics.

Given these unresolved issues, I recommend a decision leaning towards a weak reject, as the core contributions are promising but not fully substantiated.

**Claims And Evidence:**

The motivation of the task is not fully explained. The authors illustrate the necessity of multiple emotions using quotes and references. However, the argument that “novel emotion categories continuously emerge” has not been proven. Unlike the subject classification task, which has a large number of categories, the categories of emotions seem to be relatively few and fixed, such as the 27 and 28 categories in the experimental dataset of the paper. The class increment task will appear unnecessary.

**Essential References Not Discussed:**

no

**Experimental Designs Or Analyses:**

Comparative experiments are insufficient. On the one hand, among the compared methods, the latest SLCIL method is only published in 2019. More importantly, some of the SOTA MLCIL methods in 2024 have not been compared, such as CSC[1].

[1] Du K, Zhou Y, Lyu F, et al. Confidence self-calibration for multi-label class-incremental learning[C]//European Conference on Computer Vision. Cham: Springer Nature Switzerland, 2024: 234-252.

**Methods And Evaluation Criteria:**

yes

**Other Comments Or Suggestions:**

no

**Other Strengths And Weaknesses:**

Pro

-	The paper is almost well-written and well-organized.
-	The proposed method can be applied to datasets of different modalities.

Con

- The motivation of the task is not fully explained. The authors illustrate the necessity of multiple emotions using quotes and references. However, the argument that “novel emotion categories continuously emerge” has not been proven. Unlike the subject classification tas,k which has a large number of categories, the categories of emotions seem to be relatively few and fixed, such as the 27 and 28 categories in the experimental dataset of the paper. The class increment task will appear unnecessary.
- Discussion on category order. Unlike general classification tasks, the relationship between different emotion categories is very different. Therefore, different category orders in the incremental process will have an impact on performance. For example, if the old task is positive, the ability to learn negative and positive in the new task is different. Please discuss. In addition, a more reasonable performance report should be the average and standard deviation of the results under different orders.
- Stability on relation-based knowledge distillation. This module distills two teacher models at the same time. What is the relationship between the two teacher models and whether they are complementary? Related discussions are recommended. In addition, distillation with old knowledge can easily reduce the plasticity of the model and lacks the ablation of λ1.
- Comparative experiments are insufficient. On the one hand, among the compared methods, the latest SLCIL method is only published in 2019. More importantly, some of the SOTA MLCIL methods in 2024 have not been compared, such as CSC[1].

**Questions For Authors:**

see weakness

**Relation To Broader Scientific Literature:**

no

**Theoretical Claims:**

yes

---

> ### Author Rebuttal · Authors · 2025-03-31
>
> **1.The motivation of the task.**
>
> The practical significance of multi-label incremental emotion decoding can be summarized as two folds.
>
> (1)	In real world Brain-Computer Interfaces, practical applications often require dynamic adaptation to new emotion categories over time. For example, in clinical BCIs for depression monitoring, initial models may focus on basic emotions (happy/sad), but later need to incorporate nuanced states (anhedonia, emotional blunting) as therapy progresses.
>
> (2)	The rapid advancement of psychology has led to increasingly fine-grained discoveries in emotion categories. Recent study [1] has empirically identified up to 80 distinct emotional states, far exceeding traditional coarse taxonomies. Besides, emotion categories are not static but evolve with interdisciplinary findings. For example, "Bittersweetness" and "awe" were later additions to emotion frameworks.
>
> **2.Discussion on category order.**
>
> If the model initially learns exclusively positive emotions followed by negative emotions in subsequent tasks, the excessive inter-task categorical divergence may exacerbate catastrophic forgetting.
> We conducted 10 randomized shuffles of the category sequence and computed the mean and standard deviation. The experimental results are presented below:
> |Method | Brain27 B0-I9 | Brain27 B0-I3 | Brain27 B15-I3| Brain27 B15-I2 |
> |-----|-------------------|------------------|---------------------|--------------------|
> |AESL|44.8 $\pm$ 0.3|43.9 $\pm$ 0.5|41.7 $\pm$ 0.5|39.9 $\pm$ 0.5|
>
> |Method | Video27 B0-I9 | Video27 B0-I3 | Video27 B15-I3| Video27 B15-I2 |
> |-----|-------------------|------------------|---------------------|--------------------|
> |AESL|45.1 $\pm$ 0.5|47.3 $\pm$ 0.4|41.6 $\pm$ 0.3|39.5 $\pm$ 0.7|
>
> |Method | Audio28 B0-I7 | Audio28 B0-I4 | Audio26 B16-I3| Audio28 B16-I2 |
> |-----|-------------------|------------------|---------------------|--------------------|
> |AESL|49.4 $\pm$ 0.6|48.5 $\pm$ 0.3|47.9 $\pm$ 0.3|45.2 $\pm$ 0.2|
>
> **3.The relationship between the two teacher models.**
>
> The two teacher models exhibit complementary characteristics. The old-model teacher serves to mitigate the forgetting of previously learned emotion categories, while the affective-dimension teacher addresses catastrophic forgetting caused by future-missing partial label problem. Besides, relying solely on distillation from the affective dimension space would lead to suboptimal model performance due to the heterogeneity of feature representations, resulting in training difficulties.
>
> **4.Distillation with old knowledge can easily reduce the plasticity of the model and lacks the ablation of λ1.**
>
> Knowledge distillation from the old model is a widely used method to mitigate catastrophic forgetting, where the balance between plasticity and stability is primarily controlled by the regularization parameter—specifically, λ₁ in Eq(15). In the Brain27 dataset with B0-I9 setting, we fixed λ₂ = 0.5 and λ₃ = 2, and observed the impact of varying λ₁ on model performance as follows:
>
> | $\lambda_1$ | 0.2   | 0.3   | 0.4  | 0.5  | 0.6  |  0.7  |  0.8  |
> |-----------------|---------|---------|--------|-------|--------|----------|---------|
> | **mAP (%)** | 41.4 | 41.9  |42.7 | 42.9  | 43.1 |  42.6 |  42.0  |
>
> We can observe that an excessively small λ₁ leads the model to distill more information from the affective dimension space. Considering the heterogeneity of the feature space, this makes the model harder to train and results in lower accuracy. On the other hand, an overly large λ₁ makes the model more susceptible to the impact of future-missing partial label problem, thereby degrading its performance.
>
> **5.Some of the SOTA MLCIL methods in 2024 have not been compared, such as CSC.**
>
> We have conducted comparative experiments with the CSC method, and the results will be incorporated into the final version of the paper. The experimental results (Avg. Acc) are as follows:
> |Method | Brain27 B0-I9 | Brain27 B0-I3 | Brain27 B15-I3| Brain27 B15-I2 |
> |-----|-------------------|------------------|---------------------|--------------------|
> |CSC|42.1 $\pm$ 0.2|43.0 $\pm$ 0.3|41.1 $\pm$ 0.6|39.4 $\pm$ 0.4|
> |AESL|44.8 $\pm$ 0.3|43.9 $\pm$ 0.5|41.7 $\pm$ 0.5|39.9 $\pm$ 0.5|
>
> |Method | Video27 B0-I9 | Video27 B0-I3 | Video27 B15-I3| Video27 B15-I2|
> |-----|-------------------|------------------|---------------------|--------------------|
> |CSC|44.1 $\pm$ 0.7|46.8 $\pm$ 0.2|41.6 $\pm$ 0.2|38.4 $\pm$ 0.6|
> |AESL|45.1 $\pm$ 0.5|47.3 $\pm$ 0.4|41.6 $\pm$ 0.3|39.5 $\pm$ 0.7|
>
> |Method | Audio28 B0-I7 | Audio28 B0-I4 | Audio26 B16-I3| Audio28 B16-I2|
> |-----|-------------------|------------------|---------------------|--------------------|
> |CSC|47.8 $\pm$ 0.3|48.0 $\pm$ 0.7|46.9 $\pm$ 0.5|45.2 $\pm$ 0.3|
> |AESL|49.4 $\pm$ 0.6|48.5 $\pm$ 0.3|47.9 $\pm$ 0.3|45.2 $\pm$ 0.2|
>
> **References:**
>
> [1] Koide-Majima N et al. Distinct dimensions of emotion in the human brain and their representation on the cortical surface. NeuroImage, 2020

---

### Official Review · Reviewer_mBeK · 2025-03-14

**Overall Recommendation:** 3

**Summary:**

The paper proposes **EmoGrowth**, a framework addressing **multi-label fine-grained class incremental emotion decoding**. This paradigm enables models to learn **new emotion categories incrementally** while preserving the ability to recognize **multiple concurrent emotions** in dynamic real-world environments.  The **Augmented Emotional Semantics Learning (AESL)** framework introduces three innovations:
- **Emotional Relation Graph (ERG)** for label disambiguation and capturing inter-class dependencies.
- **Affective dimension-based knowledge distillation** to align features with continuous emotion representations.
- **Semantic-specific feature decoupling** guided by emotion embeddings.

Experiments on **brain activity (Brain27)**, **video (Video27)**, and **audio (Audio28)** datasets demonstrate superior performance in decoding **28 fine-grained emotions**, outperforming existing methods while mitigating catastrophic forgetting.

---

### **Key Contributions**
1. **Technical Innovation**:
   - **Augmented ERG**: Dynamically integrates historical and new task emotional relationships using graph-based label disambiguation. This resolves the *past-missing label problem* by generating reliable soft labels and preserving label correlations across tasks.
   - **Relation-based Knowledge Distillation (RKD)**: Aligns model features with **affective dimension spaces** (e.g., valence-arousal), leveraging domain knowledge to address the *future-missing label problem*.
   - **Emotional Semantics Learning**: A graph autoencoder learns emotion embeddings for semantic-guided feature decoupling, enhancing multi-label recognition.

2. **Empirical Validation**:
   - Evaluated under multiple incremental protocols (e.g., B0-I7, B16-I3), AESL achieves **state-of-the-art results**, surpassing SLCIL methods (e.g., LwF, ER) and MLCIL baselines (e.g., AGCN, OCDM) by large margins (e.g., **15–20% higher mAP** on Audio28).
   - Visualization (t-SNE, ERG adjacency matrices) validates emotion embeddings’ semantic topology and label relationship reconstruction.

---

**Claims And Evidence:**

### **Weaknesses**
1. **Limited Exploration of Task Dynamics**:
   - The impact of **emotion category ordering** (e.g., learning "Adoration" before "Awe") on performance remains unexamined, which could affect real-world deployment.
   - **Task granularity**: The effect of varying emotion categories per task (e.g., 2 vs. 7 categories per incremental task) is not analyzed.

2. **Practical Constraints**:
   - While affective dimensions (valence/arousal) are utilized, their annotation process for new tasks is not discussed—raising questions about scalability.
   - No evaluation on **cross-subject** or **cross-domain adaptation** (e.g., transferring from brain signals to audio), limiting applicability to heterogeneous data scenarios.

3. **Theoretical Gaps**:
   - The interplay between emotion categories and affective dimensions is underexplored. For instance, how explicit alignment (via RKD) improves incremental learning beyond empirical results lacks theoretical justification.

---

### **Suggestions**
1. **Expand Task Dynamics Analysis**:
   - Conduct ablation studies on emotion order and task size variability.
   - Explore **curriculum learning** strategies to optimize task sequences.

2. **Enhance Generalizability**:
   - Test cross-domain incremental learning (e.g., Brain27 → Audio28).
   - Investigate unsupervised/semi-supervised affective dimension estimation for new tasks.

3. **Deepen Theoretical Foundation**:
   - Formalize guarantees for knowledge transfer between affective spaces and emotion categories.

**Essential References Not Discussed:**

NO

**Experimental Designs Or Analyses:**

SEE Claims And Evidence

**Methods And Evaluation Criteria:**

SEE Claims And Evidence

**Other Comments Or Suggestions:**

SEE Claims And Evidence

**Other Strengths And Weaknesses:**

SEE Claims And Evidence

**Questions For Authors:**

SEE Claims And Evidence

**Relation To Broader Scientific Literature:**

SEE Claims And Evidence

**Theoretical Claims:**

SEE Claims And Evidence

---

> ### Author Rebuttal · Authors · 2025-03-31
>
> **1.Conduct ablation studies on emotion order and task size variability.**
>
> For the emotion order, we conducted 10 randomized shuffles of the category sequence. The experimental results are presented below:
> |Method | Brain27 B0-I9 | Brain27 B0-I3 | Brain27 B15-I3| Brain27 B15-I2 |
> |-----|-------------------|------------------|---------------------|--------------------|
> |AESL|44.8 $\pm$ 0.3|43.9 $\pm$ 0.5|41.7 $\pm$ 0.5|39.9 $\pm$ 0.5 |
>
> |Method | Video27 B0-I9 | Video27 B0-I3 | Video27 B15-I3| Video27 B15-I2 |
> |-----|-------------------|------------------|---------------------|--------------------|
> |AESL|45.1 $\pm$ 0.5|47.3 $\pm$ 0.4|41.6 $\pm$ 0.3|39.5 $\pm$ 0.7|
>
> |Method | Audio28 B0-I7 | Audio28 B0-I4 | Audio26 B16-I3| Audio28 B16-I2 |
> |-----|-------------------|------------------|---------------------|--------------------|
> |AESL|49.4 $\pm$ 0.6|48.5 $\pm$ 0.3|47.9 $\pm$ 0.3|45.2 $\pm$ 0.2   |
>
> For the task size variability, we have conducted experiments for different emotion categories per task. The experimental results are presented below:
>
> |Method | Brain27 B0-I9 | Brain27 B0-I7 | Brain27 B0-I5| Brain27 B0-I3 |Brain27 B0-I1|
> |-----|-------------------|------------------|---------------------|--------------------|-----|
> |AESL|44.8 $\pm$ 0.3|44.6 $\pm$ 0.4|43.9 $\pm$ 0.3|43.9 $\pm$ 0.5    |43.0$\pm$ 0.9|
>
> |Method | Video27 B0-I9 | Video27 B0-I7 | Video27 B0-I5| Video27 B0-I3 |Video27 B0-I1|
> |-----|-------------------|------------------|---------------------|--------------------|-----|
> |AESL|45.1 $\pm$ 0.5|46.8 $\pm$ 0.2|46.4 $\pm$ 0.2|47.3 $\pm$ 0.4  |44.8 $\pm$ 0.5|
>
> |Method | Audio28 B0-I7 | Audio28 B0-I5 | Audio28 B0-I4| Audio28 B0-I2 |Audio28 B0-I1|
> |-----|-------------------|------------------|---------------------|--------------------|------|
> |AESL|49.4 $\pm$ 0.6|49.0 $\pm$ 0.2|48.5 $\pm$ 0.3|47.4 $\pm$ 0.7  |46.1$\pm$ 0.9|
>
> In most cases, as the task size decreases and the number of learned tasks increases, the model performance exhibits an overall declining trend.
>
> **2.Explore curriculum learning strategies to optimize task sequences.**
>
> We appreciate the valuable suggestion regarding curriculum learning for task sequencing. However, in practical deployment scenarios, the emotion categories appearing at each incremental stage are typically unforeseen in advance. This inherent uncertainty makes our current approach of randomized task shuffling with repeated evaluations (as shown above) a methodologically sound solution, ensuring robustness to arbitrary category arrival orders.
>
> **3.Test cross-domain incremental learning (e.g., Brain27 → Audio28).**
>
> We have conducted cross-domain incremental learning experiments from Brain27 to Audio28. The results on the transferred Audio28 dataset are compared with those obtained by training directly on Audio28 from scratch as follows:
>
> |Method | Audio28 B0-I7 | Audio28 B0-I4 | Audio26 B16-I3| Audio28 B16-I2 |
> |-----|-------------------|------------------|---------------------|--------------------|
> |Brain→Audio|49.6 $\pm$ 0.1|48.4 $\pm$ 0.2|47.4 $\pm$ 0.5|45.5 $\pm$ 0.3 |
> |Audio|49.4 $\pm$ 0.6|48.5 $\pm$ 0.3|47.9 $\pm$ 0.3|45.2 $\pm$ 0.2|
>
> The experimental results demonstrate that transfer learning can enhance model performance compared to cold-start training in certain scenarios. However, due to the significant feature disparity, the cross-domain transfer may conversely degrade performance in some cases.
>
> **4.Investigate unsupervised/semi-supervised affective dimension estimation for new tasks.**
>
> In our work, we assume the affective dimension labels of samples are manually annotated ratings. Considering the prohibitive cost of manually rating each new sample in practical applications, we explored pre-training an affective dimension scoring model with additional held-out training set. This allows us to automatically extract affective dimension features for incoming samples using the pre-trained model. Comparative experimental results between knowledge distillation using model-predicted affective dimension labels versus human-annotated labels are presented below:
>
> |Method | Brain27 B0-I9 | Brain27 B0-I3 | Brain27 B15-I3| Brain27 B15-I2 |
> |-----|-------------------|------------------|---------------------|--------------------|
> |Model-predicted|44.5 $\pm$ 0.2|43.8 $\pm$ 0.5|41.7 $\pm$ 0.4|39.7 $\pm$ 0.3  |
> |Human-annotated |44.8 $\pm$ 0.3|43.9 $\pm$ 0.5|41.7 $\pm$ 0.5|39.9 $\pm$ 0.5 |
>
> The results demonstrate that utilizing model-predicted affective dimension labels can achieve comparable performance to using human-annotated affective dimension labels.
>
> **5.Formalize guarantees for knowledge transfer between affective spaces and emotion categories.**
>
> We sincerely appreciate this foundational question. In our upcoming work, we will establish error bounds for cross-space knowledge transfer between emotion category manifolds and dimensional affective spaces by deriving Lipschitz continuity conditions and optimal transport-based projection guarantees.

---

### Decision · Program_Chairs · 2025-05-01

**Decision:**

Accept (poster)

**Comment:**

The paper received one weak reject and three weak accepts. Some weaknesses include limited exploration of task dynamics, motivation needs more details, and the significance of the problem needs better detailed. Strengths include nice technical innovation, paper is well written, and experiments are well designed, the problem is interesting, and the proposed approach has good novelty. Considering this, the strengths outweigh the weaknesses.